# Chromatin accessibility landscapes of skin cells in systemic sclerosis nominate dendritic cells in disease pathogenesis

Qian Liu[1,8], Lisa Zaba[2,3,8], Ansuman T. Satpathy[2], Michelle Longmire[2,3], Wen Zhang[1], Kun Li[1], Jeffrey Granja[2,3], Chuang Guo[1], Jun Lin[1], Rui Li[2,3], Karen Tolentino[2,3], Gabriela Kania[4], Oliver Distler[4], David Fiorentino[3], Lorinda Chung[3,5], Kun Qu[1,6,7✉] & Howard Y. Chang[2,3✉]

Systemic sclerosis (SSc) is a disease at the intersection of autoimmunity and fibrosis. However, the epigenetic regulation and the contributions of diverse cell types to SSc remain unclear. Here we survey, using ATAC-seq, the active DNA regulatory elements of eight types of primary cells in normal skin from healthy controls, as well as clinically affected and unaffected skin from SSc patients. We find that accessible DNA elements in skin-resident dendritic cells (DCs) exhibit the highest enrichment of SSc-associated single-nucleotide polymorphisms (SNPs) and predict the degrees of skin fibrosis in patients. DCs also have the greatest disease-associated changes in chromatin accessibility and the strongest alteration of cell–cell interactions in SSc lesions. Lastly, data from an independent cohort of patients with SSc confirm a significant increase of DCs in lesioned skin. Thus, the DCs epigenome links inherited susceptibility and clinically apparent fibrosis in SSc skin, and can be an important driver of SSc pathogenesis.

[1]Department of Oncology, The First Affiliated Hospital of USTC, Division of Molecular Medicine, Hefei National Laboratory for Physical Sciences at Microscale, the CAS Key Laboratory of Innate Immunity and Chronic Disease, Division of Life Sciences and Medicine, University of Science and Technology of China, Hefei 230021, China. [2]Center for Personal Dynamic Regulomes, Stanford University School of Medicine, Stanford CA, USA. [3]Department of Dermatology, Stanford University School of Medicine, Stanford, CA 94305, USA. [4]Department of Rheumatology, University Hospital Zurich, Zurich, Switzerland. [5]Division of Rheumatology, Department of Medicine, Stanford University School of Medicine, Stanford, CA 94305, USA. [6]CAS Center for Excellence in Molecular Cell Sciences, University of Science and Technology of China, Hefei 230027, China. [7]School of Data Sciences, University of Science and Technology of China, Hefei 230027, China. [8]These authors contributed equally: Qian Liu, Lisa Zaba. ✉email: qukun@ustc.edu.cn; howchang@stanford.edu

Systemic sclerosis (SSc), also known as scleroderma, is a chronic multi-system disease that is characterized by vascular damage, inflammation, and progressive fibrosis of the skin and internal organs. Almost all patients with SSc have skin involvement despite the heterogeneity of the disease, and fibrosis leads to internal organ dysfunction that is the most common cause of death in these patients[1,2]. Tightened and thickened skin is the clinical hallmark of SSc, and is often associated with distinct patterns of organ involvement, disease severity, and survival[3]. Previous studies focused on identifying the signaling pathways that were involved in fibrosis of the internal organs and ultimately to reduce the mortality rate of SSc[4]. Current studies indicate that genetic predisposition combined with environmental triggers, such as chronic tissue damage, vascular insult, or incipient cancer can lead to local inflammatory microenvironments[1,5]. However, the epigenetic regulatory mechanism of SSc and how the immune cells in the skin microenvironment contribute to the disease remain largely unknown.

Because only 1% of the human genome is accessible in any given cell type, the identity and pattern of accessible DNA is highly informative of cell states and regulatory programs[6]. Recent innovations in epigenetic profiling have led to the development of Assay of Transposase Accessible Chromatin with sequencing (ATAC-seq), enabling direct and sensitive detection of open chromatin regions and generation of high-resolution chromatin maps from as few as 500 cells[7], or even in single cells[8,9]. ATAC-seq has been widely applied in many biomedical systems, such as cancer and immunity, to study the epigenetic landscapes and regulomes that drive disease pathogenesis in vivo[6,10–12].

Here we survey the genome-wide active regulatory elements in fresh human skin cells in vivo, by performing ATAC-seq on skin samples from healthy volunteers, as well as affected and unaffected skin from patients with SSc. We create high-resolution epigenetic regulomes of multiple cell types in normal and SSc skin samples, and perform in-depth analysis of the regulatory mechanisms of the disease. Our results suggest that DCs display the strongest correlation with skin fibrosis and the greatest alteration of epigenome than the other cell types. Our study thus provides a better understanding of the functions of DCs in driving SSc and a rich source of candidates for therapeutic targets to treat the disease.

## Results

**Chromatin accessibility landscapes of 8 cell types from normal human skin in vivo.** To establish a baseline normal chromatin landscape, we first harvested cells directly from fresh human skin and analyzed the genome-wide chromatin accessibility maps of 19 samples from 8 cell types resident in the skin, including CD4+ and CD8+ T cells (CD4s, CD8s), dendritic cells (DCs), Langerhans (LCs), endothelial cells (ECs), macrophages (Macs), fibroblasts (Fibs), and keratinocytes (KCs) (Fig. 1a, Supplementary Fig. 1a, b, Supplementary Data 1-2, see Methods). Each ATAC-seq library was sequenced to obtain an average of more than 25 million paired-end reads, in total comprising over 500 million measurements (Supplementary Data 3). We used a published ATAC-seq pipeline[13] to analyze raw sequencing data and identify focal peaks of chromatin accessibility that typify active regulatory elements. After filtering and quantile normalization, we identified a total of 104,223 high-quality accessible elements across these 8 skin resident cell types.

Transcription start site (TSS) enrichment and read length distribution analysis of all normal samples demonstrated the high quality of the dataset (Supplementary Fig. 1c−d), and the Pearson correlation coefficients of all the samples suggested excellent reproducibility between the biological replicates of most individual cell types (Supplementary Fig. 1e). For each cell type, ATAC-seq successfully detected open chromatin signals around lineage-specific marker genes (Supplementary Fig. 1f). A snapshot of the ATAC-seq profiles indicated high signal-to-noise ratio of these data, capturing the known enhancer and promoter elements previously identified by histone H3 lysine 27 acetylation chromatin immunoprecipitation sequencing in a large compendium of cells surveyed by the ENCODE project (Fig. 1b).

Since the regulatory elements in skin biopsies and cells from in vitro expansion are quite different[14], we sought to quantify the potential differences in the chromatin landscape of cells directly harvested from fresh skin compared to cells from tissue culture. Take fibroblasts as an example, we found 12768 accessible elements (over 12% of all detected accessible sites) were significantly differential ($|\log_2$ Fold change$| > 4$, $P$ value $< 0.05$) (Supplementary Fig. 2a–c), indicating that the native milieu of skin cells does differ from that of skin cells in culture at the chromatin level. Similar results were also obtained in KCs, where 8% of detected peaks in KCs from skin biopsy were found significant differential ($|\log_2$ Fold change$| > 4$, $P$ value $< 0.05$) from that of the cultured cells (Supplementary Fig. 2d–e).

As distal enhancers (peaks>1 kb away from the closest TSS) provide significantly improved cell type classification compared to promoters and transcription profiles[15], we then performed unsupervised clustering and principal component analysis based on chromatin accessibilities of the distal enhancers for normal samples, and found that all the samples were precisely classified into each individual cell type (Fig. 1c, Supplementary Fig. 1g), confirming the high quality of the dataset. Furthermore, our results suggested the similarity of the chromatin open state of dermal macrophages and CD31+ endothelial cells. The correlation analysis across different cell types from SSc patients also showed a strong correlation between Macs and ECs (Supplementary Fig. 3a). Since dermal Macs and DCs were both differentiated from monocytes, our normalized ATAC-seq profiles showed that chromatin around several marker genes of myeloid cells, such as *ITGAX* (CD11C), *CD80*, *CD68*, *HLA-DRA*, *TLR4* were indeed more accessible in Macs and DCs than other cell types (Supplementary Fig. 3b), indicating the reliability of our ATAC-seq data of macrophage. Thus, we obtained the first reliable chromatin accessibility profiles of multiple rare skin cell types harvested directly from human skin in vivo, which are critical for physiologically relevant downstream analysis, highlighting the value of our study.

**Cell type-specific chromatin accessibility in normal skin.** We next explored the regulomes of different cell types in normal skin from healthy individuals. Peaks from each cell type were compared with the remaining samples, and a total of 14243 significant cell-type-specific peaks were identified ($P$ value $< 0.005$, $|\log_2$ Fold change$| > 2$, Fig. 2a). These peaks were then clustered into 5 major groups representing the specific chromatin accessible sites of T cells, DCs and LCs, Fibs, ECs and Macs, and KCs. Peaks around functional marker genes for each cell type were also shown. Non-repetitive top enriched genes (Supplementary Data 4) and biological functions (Supplementary Data 5) for each cluster were annotated using the genomic regions enrichment of annotations tool (GREAT)[16] (Fig. 2b). We noted that these top enriched GO terms for each peak cluster were also consistent with the identities of each cell group. For example, T cell-specific peaks were enriched with biological functions such as regulation of immune system processes ($P$ value $< 10^{-81}$) and leukocyte activation ($P$ value $< 10^{-78}$), while KC-specific peaks were enriched for keratinocyte differentiation ($P$ value $< 10^{-13}$).

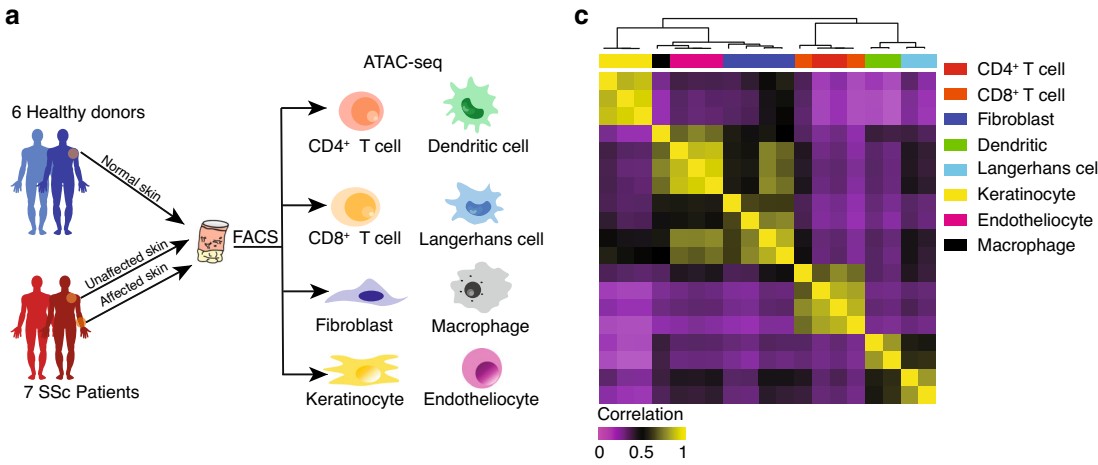

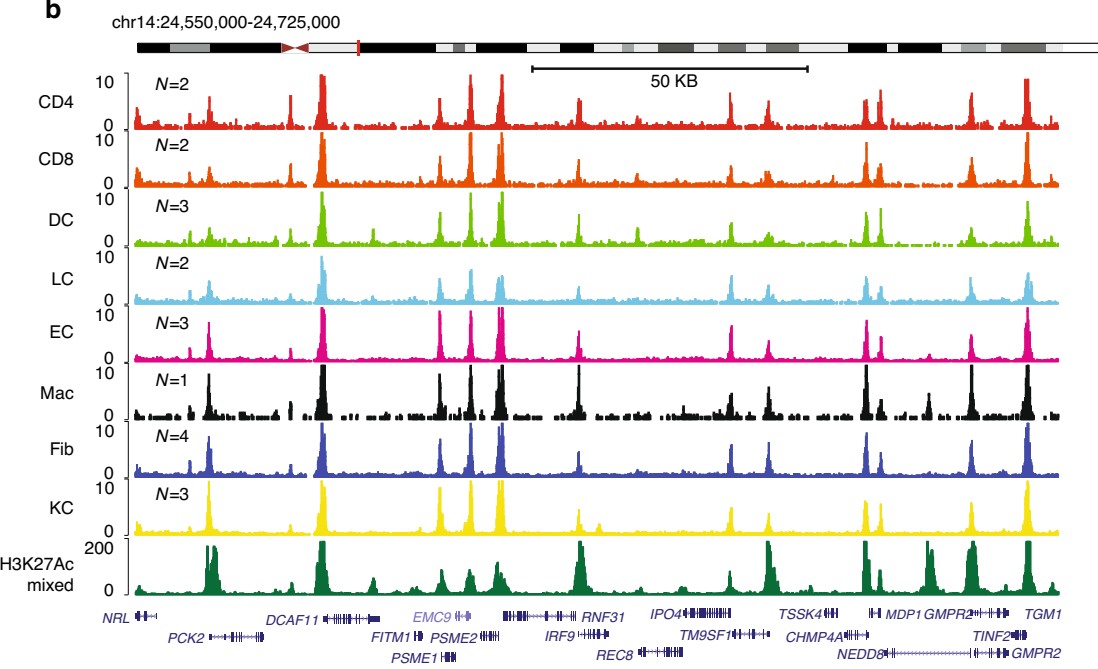

**Fig. 1 Landscape of DNA accessibility in 8 cell types from normal skin in vivo.** (**a**) A schematic outline of the study design depicting the workflow for the isolation and ATAC-seq of 8 cell types (CD4+, CD8+ T cells (CD4, CD8), dendritic cells (DC), Langerhans cells (LC), endotheliocytes (EC), macrophages (Mac), fibroblasts (Fib), and keratinocytes (KC)) from healthy individuals (normal skin) and SSc patients (unaffected skin and affected skin). (**b**) Normalized ATAC-seq signal profiles at a locus in CD4, CD8, DC, LC, EC, Mac, Fib, and KC from healthy donors, shown together with a normalized H2K27ac chromatin immunoprecipitation sequencing profile. *N* represents the number of biological replicates. (**c**) Unsupervised hierarchical clustering of the Pearson correlations between all the samples. ATAC-seq signals were obtained from distal elements. Each row and each column is a sample, and cell types distinguished colors. Source data are provided as a Source Data file.

Since the chromatin accessible patterns of dermal macrophages and CD31+ endothelial cells are similar, signature peaks of macrophages and endothelial cells were both enriched the biological functions about angiogenesis and wound healing (Fig. 2b, Supplementary Fig. 4a, b). Macrophages are very plastic cells, and one aspect of its heterogeneity is the tissue specialization of resident macrophages[17]. The dermal macrophages have been reported involved in angiogenesis through the expression of vascular growth factor[18]. Our results further suggested that the epigenetic regulome of macrophage residing in the dermal layer is very different from that of other myeloid cells but similar to that of endothelial cells.

Because transcription factors (TFs) bind to their cognate DNA sequences (termed motifs), by integrating the known TF motifs with DNA accessibility data from ATAC-seq, we can predict the regulome of each cell type[6]. We first obtained a total of 242 vertebrate TF motifs from the Jaspear database[19], identified their genome-wide distribution using HOMER[20], and overlaid these sites with the differential ATAC-seq peaks shown in Fig. 2a. We then used Genomica[21] to identify motifs that were significantly enriched or depleted in each sample, and thereby constructed the TF regulomes of each cell type (Fig. 2c, Supplementary Data 6). We found that most cell-type-specific TFs were consistently enriched with their corresponding cell type from our data. For instance, RUNX and GATA1, two known T cell regulators[22], were significantly enriched in CD4+ and CD8+ T cells. IRF and NFκB, critical TFs for DC function and development[23], were strongly enriched in DC-specific peaks. As another example, TFs such as TP63, KLF4, and GRHL2, which has been well-characterized in regulating KC differentiation, were found

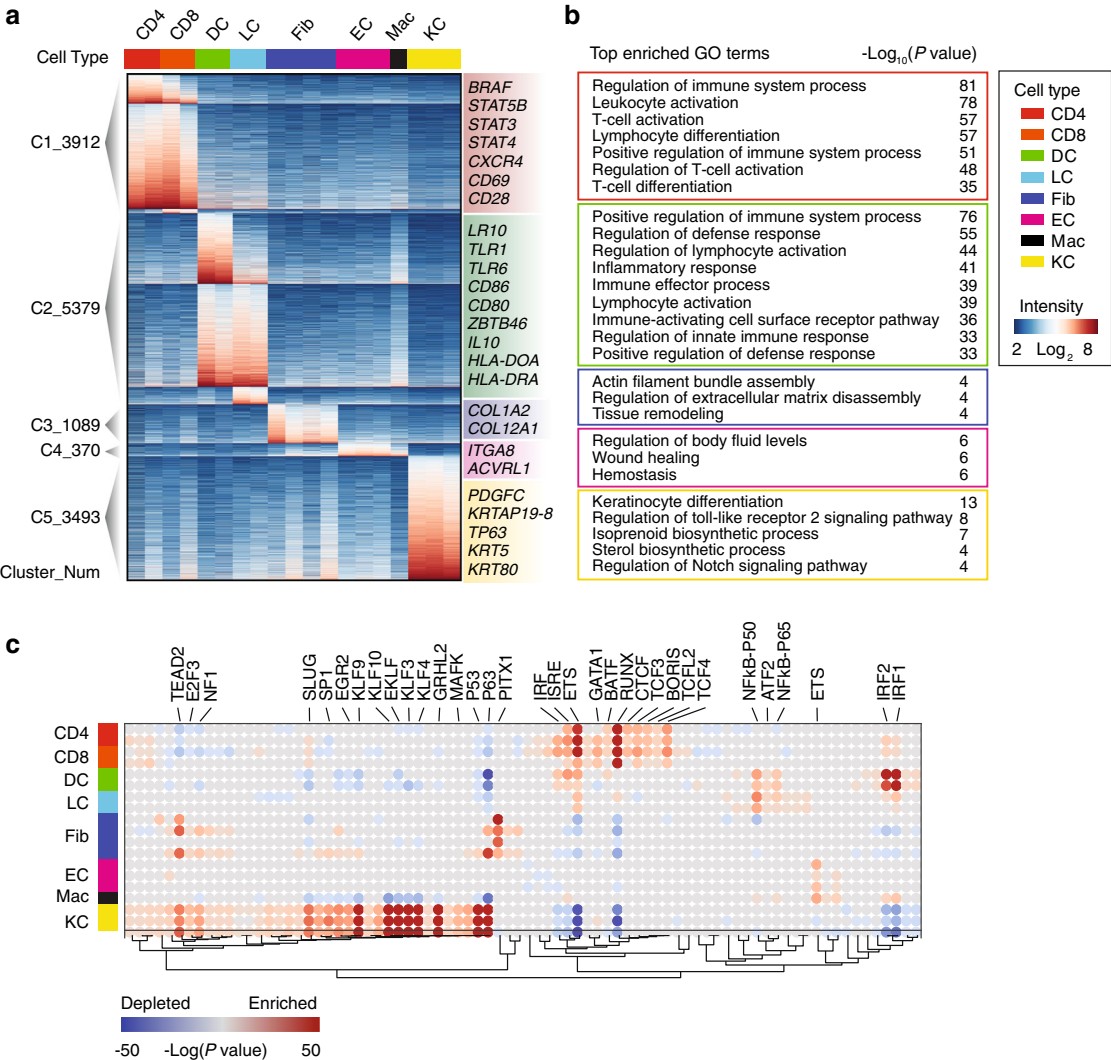

**Fig. 2 Cell type-specific chromatin accessibility in a skin biopsy from healthy donors.** (**a**) Heatmap of the normalized ATAC-seq intensities of cell type-specific peaks from healthy donors. Each row is a peak, and each column is a sample, with color-coded cell types (top panel). Clusters shown in the sidebar represent cell-type-specific peaks of CD4 (CD4+ T cells) and CD8 (CD8+ T cells) (C1), DC (dendritic cells) and LC (Langerhans cells) (C2), Fib (fibroblasts) (C3), EC (endotheliocytes) and Mac (macrophages) (C4), and KC (keratinocytes) (C5) respectively. Functional marker genes in each cluster were shown on the right. Source data are provided as a Source Data file. (**b**) Top enriched GO (Gene Ontology) terms of peaks in each cluster. P values (Binom Raw P value) were calculated using the binomial statistic test in GREAT. Source data are provided as a Source Data file. (**c**) Enrichment of known transcription factor (TF) motifs in cell type-specific accessible elements for all normal samples. Each row is a TF motif and each column is a sample. The color bar represents the significance of enrichment estimated from Genomica, where red indicates enriched and blue depleted. Source data are provided as a Source Data file.

enriched in KC-specific peaks[24]. Very few TFs were found enriched in multiple cell types, such as the TEAD2 and ETS family. These results provide a comprehensive picture of regulomes for most of the skin resident cells in vivo.

A recent study has shown that disease-associated single-nucleotide polymorphisms (SNPs) were highly enriched in noncoding DNA regulatory elements characterized by accessible chromatin[15]. By measuring the activity of regulatory elements that overlap regions with associated functional variation from GWAS, it is now possible to more accurately predict the specific cell type that is affected by genetic variants linked to diverse human diseases[15]. We then applied this method to predict the possible pathogenic cell types for SSc based on SNPs associated with immunological and skin-related diseases (Supplementary Data 7). As expected, SNPs associated with ulcerative colitis, celiac disease, and Crohn's disease were enriched in T cell-specific chromatin accessible sites (Supplementary Fig. 5a, Supplementary

Data 8), consistent with results from previous studies[11,25]. Intriguingly, SSc-associated SNPs were predominantly enriched in DCs, especially at strong associations with the highest level of statistical significance (P value < $10^{-5}$, Supplementary Fig. 5a, b), indicating that DCs may contribute to this disease through an unknown mechanism.

To further evaluate the pathogenic effects of different cell types to SSc, we downloaded the published microarray gene expression data of SSc affected skin (a total of 105 arm samples obtained from 30 patients) at 3-4 time points along the treatment of mycophenolate mofetil (MMF)[26] and performed a correlation analysis of the average expressions of cell type-specific genes (Supplementary Data 4) versus the degrees of fibrosis of the skin from SSc patients, measured by the modified Rodnan skin scores (mRSS)[27]. To remove the impact of MMF treatment on the correlation analysis, genes response to the MMF treatment (Supplementary Fig. 6a, Supplementary Data 9) were removed

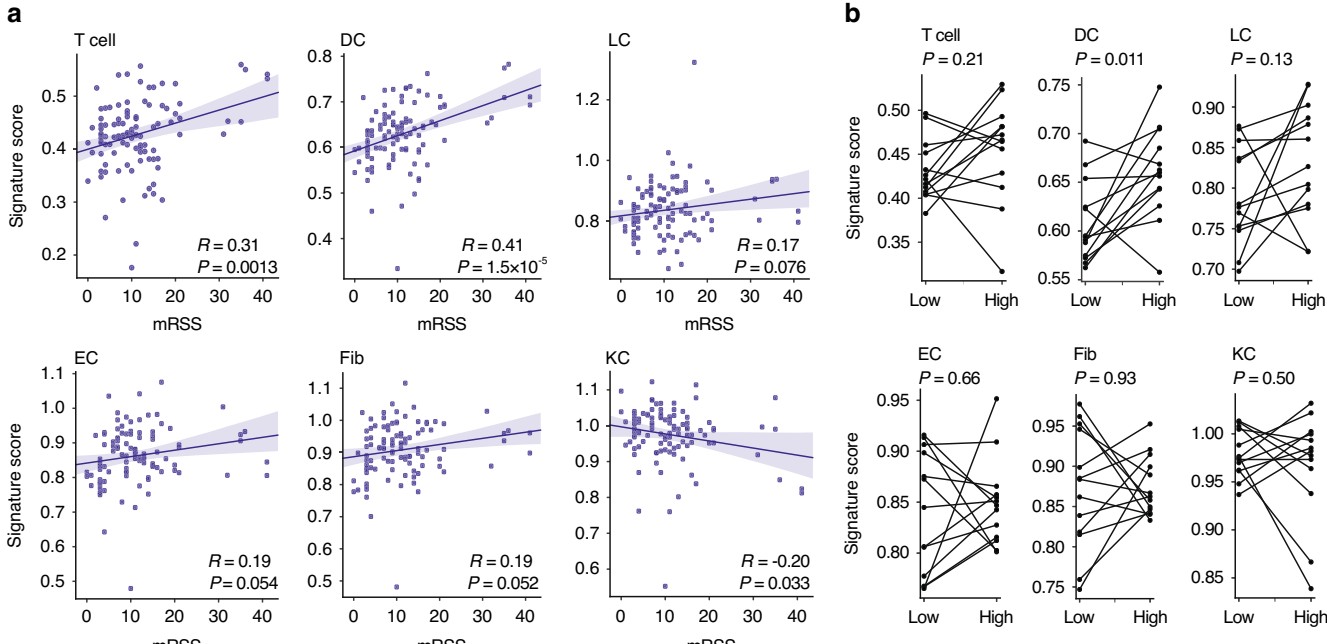

**Fig. 3 Pathogenic effects of different cell types to SSc. (a)** Scatter plots showing the average expressions of the cell type signature genes (defined in HOMER) for T cells (CD4$^+$ and CD8$^+$ T cells), DC (dendritic cells), LC (Langerhans cells), EC (endotheliocytes), Fib (fibroblasts), KC (keratinocytes), versus the corresponding mRSS (modified Rodnan skin score) of the same patient at the same time point during the treatment. Gene expression profiles were obtained from patients' affected skin cells via microarray, and genes responding to mycophenolate mofetil (MMF) treatment were discarded. The linear regression curve and 95% confidence interval (grey area) were also illustrated in each panel, two-tailed t-statistic $P$-value and coefficient ($R$) of Pearson's correlation were shown in the bottom right. Source data are provided as a Source Data file. **(b)** Pair-wise comparison of the average expressions of cell type signature genes at time point with the lowest versus highest mRSS of the same patient for all the six-cell types measured. For each patient during the treatment, we identified the time points when the mRSS score is the lowest (time point Low) and highest (time point High), and then calculated the corresponding signature scores for each cell type from the microarray profiles at time point Low and time point High, respectively. The time points Low and High can be different for different patients. 13 patients whose highest mRSS - lowest mRSS > 5 were shown, $P$ values were estimated by paired and two-tailed Student's t test. Source data are provided as a Source Data file.

from the input gene list before the correlation analysis was performed (see Methods). We found that the average expressions of DC signature genes were most significantly positively correlated with mRSS ($P$ value = $1.5 \times 10^{-5}$, $R = 0.41$, Fig. 3a, Supplementary Data 10) among the six-cell types examined, suggesting that DC was the most relevant cell type to disease pathology. A T cells gene signature was also highly correlated with mRSS ($P$ value = 0.0013, $R = 0.31$), while those of the other cell types, such as Fib, LC, EC, and KC were not significant ($P$ value > 0.05). We further performed a pair-wise comparison of cell signature gene expressions obtained at the time point when the mRSS of the same patient is at the lowest (Low) versus the highest (High), and we found that the expressions of DC signature genes were significantly higher in skin biopsy at higher mRSS levels (Student's paired two-tailed t-test $P$ value = 0.011, Fig. 3b). Other cell types, however, showed no significant differences. These results suggested that DCs may be a critical contributor of SSc.

**Cell type-specific regulome divergence in normal, unaffected, and affected skin**. To further illustrate the regulome divergence of skin from normal and SSc patients, we followed the experimental design first employed by Whitfield et al.[28], and obtained biopsies of clinically affected skin from the distal forearms and clinically unaffected skin from the lower backs of SSc patients (see Methods). Each skin sample was subjected to FACS to isolate distinct cell types followed by ATAC-seq (Supplementary Fig. 1b, Supplementary Data 1–3). Principal component analysis of all the samples clearly separated major cell types, and each individual

clinical state was further distinguished in DCs (Supplementary Fig. 7a).

To investigate the differences in chromatin accessibility and identify the epigenetic signatures that underlie SSc, we performed a pair-wise comparison of ATAC-seq profiles between clinically affected vs. unaffected vs. normal skin biopsies on T cells, DCs and fibroblasts, for which sufficient biological replicates were sequenced. Using a similar confidence level ($P$ value < 0.01, | $\log_2$ Fold change | > 2), we identified significantly more differential peaks in DCs (15869) than CD4$^+$ T cells (3786), CD8$^+$ T cells (3048), and fibroblasts (2179), suggesting that DCs may bear the most chromatin divergence between healthy and disease states (Fig. 4a–d). The differential peaks were then grouped into 6 clusters depending on their enrichment in normal, unaffected and affected cells, and the proportions of each cluster versus the differential peaks in each cell type were also evaluated (Fig. 4e–h, Supplementary Fig. 7b, Supplementary Data 11). For macrophage and EC, where only 1 sample in normal control or unaffected and affected skin were obtained, we thereby were unable to screen out the cell type-specific signature peaks with statistical power. Differential peaks of these two cell types were then defined by | $\log_2$ Fold change | > 2, however the up-regulated peaks in affected macrophage and EC were not enriched to any autoimmune fibrosis relevant GO terms (Supplementary Fig. 7c, d).

To further investigate the disease relevance and biological functions of these differential peaks, we performed disease and gene ontology analysis of all the peaks in each cluster for each cell type. We found: (1) Peaks in cluster 5, which were highly enriched in affected cells compared with normal and unaffected

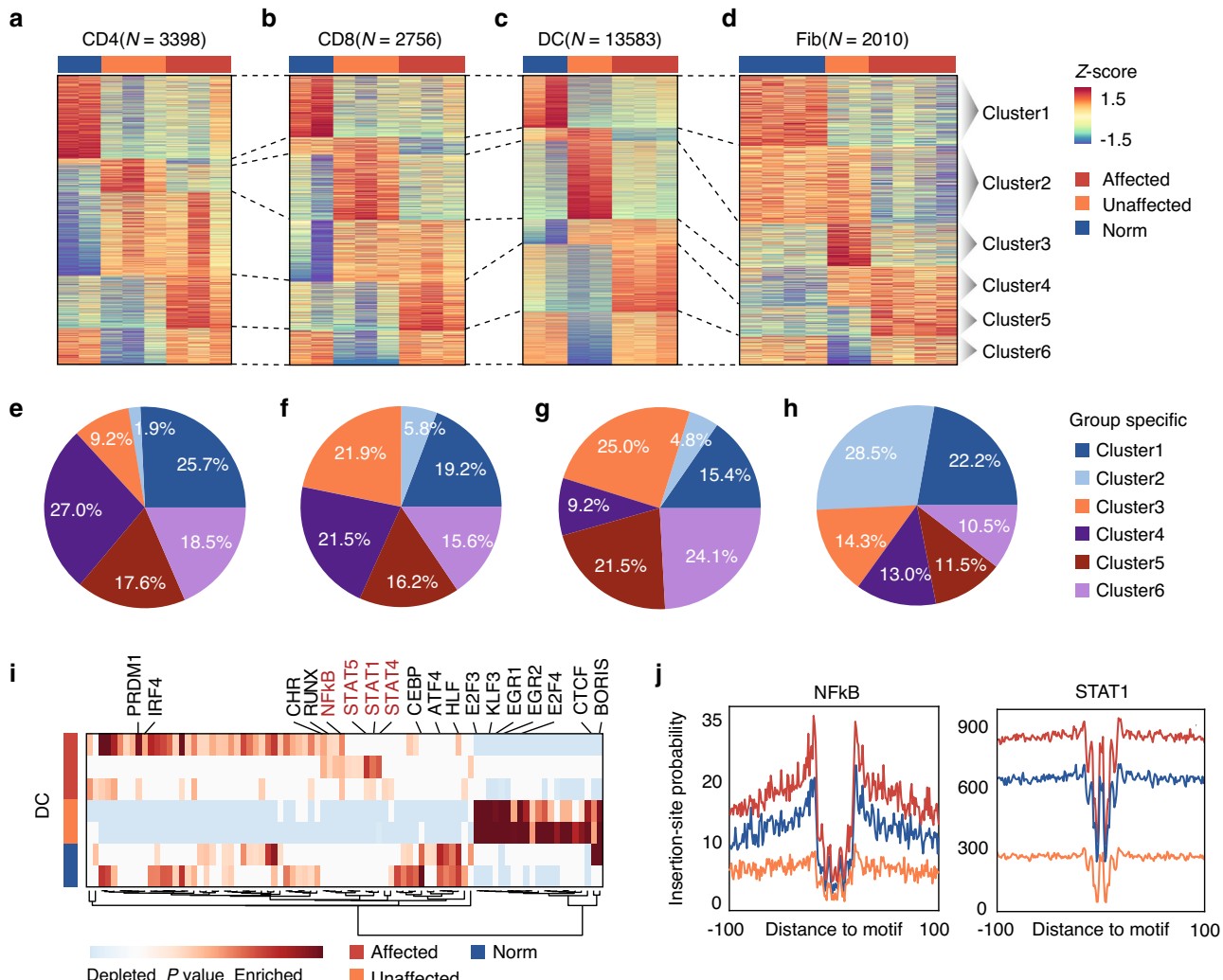

**Fig. 4 Cell types-specific regulome divergence in normal, unaffected, and affected skins.** (**a–d**) Heatmaps of the normalized ATAC-seq intensities (z-scores) of peaks enriched in normal, unaffected and affected CD4 (CD4+ T cells) (**a**), CD8 (CD8+ T cells) (**b**), DC (dendritic cells) (**c**) and Fib (fibroblasts) (**d**). Cluster 1-6 represents the peak groups enriched in normal only, normal and unaffected, unaffected only, unaffected and affected, affected only, and normal and affected cells respectively. Each row is a peak and each column is a sample. Source data are provided as a Source Data file. (**e–h**) Ratios of peaks in each cluster compare with total number of significant differential peaks in CD4 (**e**), CD8 (**f**), DC (**g**), and Fib (**h**). (**i**) Enrichment of known transcription factor (TF) motifs in DCs isolated from healthy donors, and unaffected and affected skins from SSc patients. Source data are provided as a Source Data file. (**j**) Comparison of aggregate footprints for NFκB and STAT1 in DCs isolated from normal, unaffected or affected skins.

cells, representing an SSc disease signature. A number of autoimmune diseases, including SSc, were significantly more enriched in these peaks in DC ($P$ value ~$10^{-14}$) compare with T cells and fibroblasts ($P$ value ~ 1, Supplementary Fig. 8, Supplementary Data 12), samλas immune relevant biological functions (Supplementary Fig. 9), indicating a hidden epigenetic divergence in DCs that may be an underestimated factor in driving SSc. (2) Peaks in cluster 4 were more accessible in SSc patients compared with healthy donors, representing a patient signature. Disease associated biological functions such as "Cellular response to TGFβ stimulus", "αβ T cell activation", "Inflammatory response" were found significantly enriched in cluster 4 peaks in T cells ($P$ value ~ $10^{-5}$, Supplementary Fig. 9a, b), suggesting that the chromatin states of the dermal T cells of SSc patients retain inherent abnormalities whether they are in the lesion or not.

We then sought to illustrate the enriched TFs that regulate each cluster in each cell type using the Genomica's module map algorithm and TF motif analysis[6]. To our surprise, we observed a

disease-specific TF regulatory pattern exclusively in the differential peaks in DCs, but not in other cell types. The NFκB and the STAT family TFs, which were reported to play central roles in autoimmune disease[29,30], were highly enriched in the affected DCs compared with the normal and unaffected DCs (Fig. 4i). A "TF footprint" analysis of our ATAC-seq profiles revealed distinct TFs NFκB and STAT footprints on genomic DNA directly from clinical SSc affected DCs versus normal cells (Fig. 4j), suggesting a more pronounced DNA occupancy of NFκB and STAT1 in DCs from affected SSc skin than their normal counterpart. These results suggest that DC is the most differential cell type among three states and may drive SSc through NFκB and STAT1 signaling pathways. To further investigate which DC subtype(s) may contribute to the disease, we mapped the disease specific chromatin accessible sites to the comprehensive single cell ATAC-seq map of human immune cells that was recently published[31]. We found that disease signature peaks were highly enriched in conventional dendritic cells (cDC), but less so in plasmacytoid dendritic cells or other myeloid cells

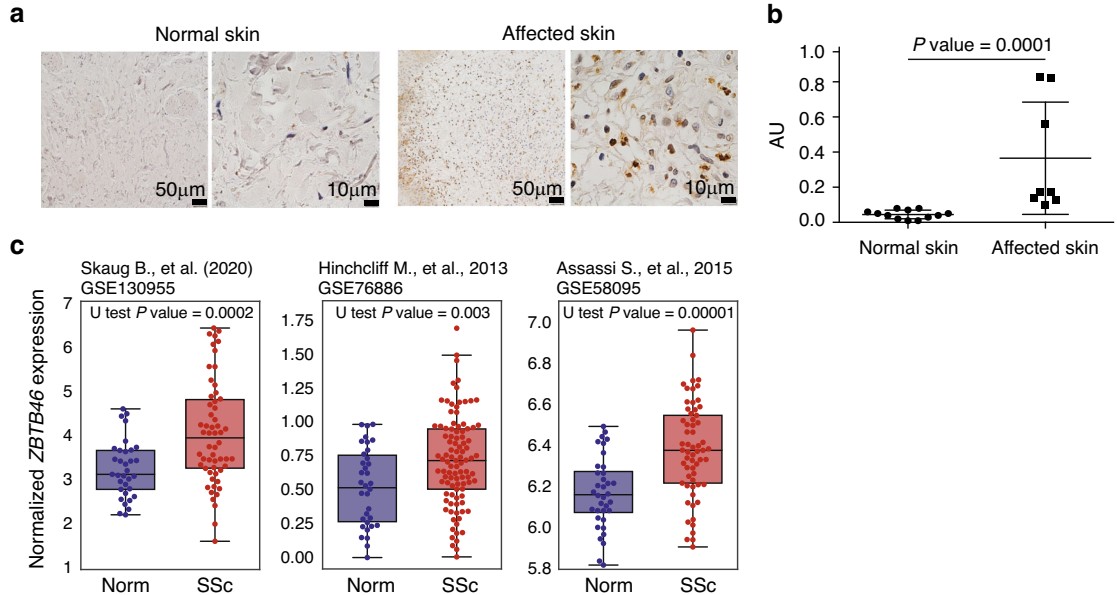

**Fig. 5 Conventional dendritic cells were more infiltrated in affected compare with normal skin.** (**a**) Immunohistochemistry analysis of ZBTB46 expression in normal and affected skin from SSc patient. 12 normal skin samples and 8 affected skin samples were included in our experiments. (**b**) Average expression of ZBTB46 protein in normal ($n = 12$) and affected ($n = 8$) skin from SSc patient. P value was estimated by one-sided Mann–Whitney U test. The upper, centre, and lower line indicates 75% quantile $+1.5$ * interquartile range (IQR), 50% quantile and 25% quantile $-1.5$ * IQR respectively. Source data are provided as a Source Data file. (**c**) The expression of cDC (conventional dendritic cells) marker gene ZBTB46 in normal ($n = 33, 34, 36$) and SSc ($n = 58, 99, 61$) skin in three published datasets. P value was estimated by one-sided Mann–Whitney U test. The upper, centre, and lower line of the box indicates 75%, 50%, and 25% quantile, respectively. The upper and lower whisker of the boxplot indicates 75% quantile $+1.5$ * interquartile range (IQR) and 25% quantile $-1.5$ * IQR. Source data are provided as a Source Data file.

(Supplementary Fig. 10a–c), suggesting that cDC was the main pathogenic subtype of dendritic cells.

We next tested whether DCs were more infiltrated in affected compare with normal skin. To do so, we examined the expression of ZBTB46 in normal skin from an independent set of healthy donors and lesional skin from SSc patients. ZBTB46 is a well-known TF selectively expressed in classical DCs and their committed progenitors but not by plasmacytoid DCs, monocytes, macrophages, or other lymphoid or myeloid lineages[32] (Supplementary Fig. 11). Importantly, there were significantly more DCs stained by ZBTB46 in affected SSc skin compared to normal skin (Fig. 5a, b, P value = 0.0001, one-sided Mann-Whitney U test). Gene expression profiles from multiple published studies[26,33,34] have also shown that gene ZBTB46 were significantly higher expressed in SSc skin versus normal skin (Fig. 5c, P value = 0.0002, 0.003, 0.00001, one-sided Mann-Whitney U test). These results further support our hypothesis that DCs play an important role in the formation of an abnormal inflammatory immune environment for fibrosis.

**DCs play a central role in communication between skin resident cells in SSc.** Reciprocal communication between divergent skin cell populations plays a key role in skin development, homeostasis and repair. Disrupted cross-talks between cells however, may cause skin fibrosis leading to SSc[35,36]. As such, we sought to illustrate the interactions between resident skin cells based on ATAC-seq data, and elucidate the signaling pathways through which connections were altered in SSc (see Methods). We first surveyed the chromatin accessibilities of several well-studied signaling pathways as positive controls, such as NOTCH and TGFβ which interconnect 4 major cell types in skin and were also known to associate with tissue fibrosis[37,38] (Fig. 6a). We noticed that the chromatin of several NOTCH and TGFβ receptors/ligands' gene loci were more accessible in SSc,

indicating an up-regulation of these pathways in a disease state between resident skin cells. We then use a circos plot to show the ATAC-seq signals around selected receptors/ligands in normal versus affected skin from SSc patients, where the outermost torus displays the names of the receptors and ligands in each cell type and the middle and inner torus display the ATAC-seq signals at these genes' loci in SSc patients and healthy controls respectively (Fig. 6b). More specifically, ATAC-seq peaks around NOTCH1 receptor were generally more accessible in affected DCs, and those of NOTCH1's ligand DLL4 were more accessible in affected CD4$^+$ T cells, suggesting the connections between DCs and CD4$^+$ T cells may become hyperactive through the NOTCH1/DLL4 pathway in SSc. Similarly, this overstimulated communication between DCs and CD4$^+$ T cells can also be achieved through costimulatory signaling (CD86/CD28) and fibrosis-associated signaling (TGFBR2/TGFB1) pathways[39].

In this way, we can nominate the potential communications between T cells, DCs and Fibs in SSc from limited number of patients' samples. We first obtained all the known receptor/ligand pairs from the Ligand-Receptor Partners (DLRP) database[40] and CellPhoneDB[41], and then asked whether the chromatin around their encoding genes loci were significantly more/less accessible in disease states. We defined a Strength of Interaction Alteration (SIA) score for each receptor/ligand pair, in which a higher SIA value denotes a stronger cell-cell connection in SSc (see Methods). We identified 80 pairs of significant altered cross-talks between these cells (Fig. 6c, Supplementary Data 13, |SIA| > 7). Our results indicated pervasive intercellular communications altered between skin resident cells through multiple diverse pathways in SSc (Supplementary Fig. 12). For instance, there were receptors/ligands significantly altered in almost all cell–cell communications in the disease state, such as the down-regulated EFNA5/EPHA4 (Fig. 6d) and up-regulated IL10/IL10RA interleukin

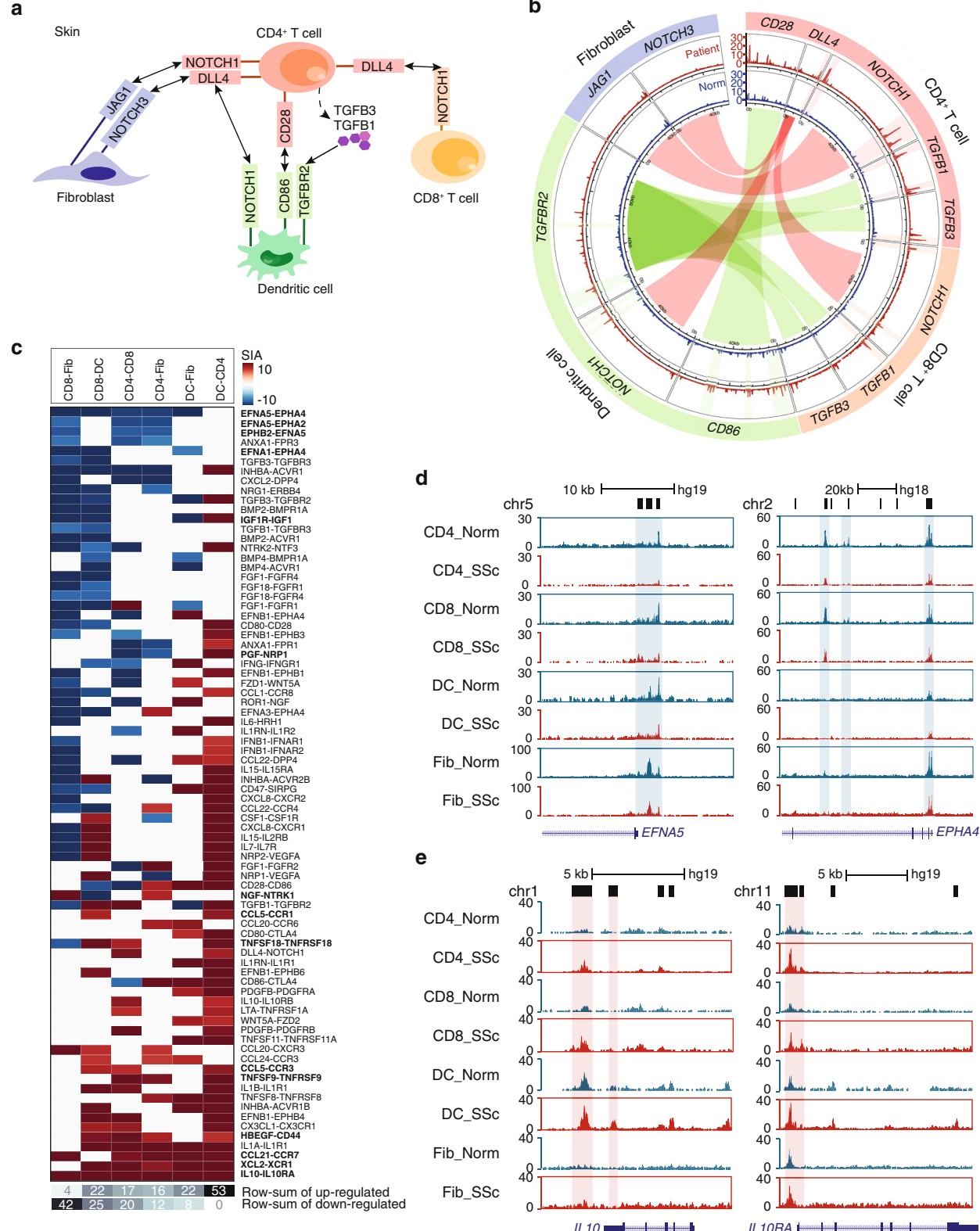

signaling pathways (Fig. 6e), whose dysregulation may increase risk for many autoimmune diseases[42,43], yet the epigenetic mechanism of these pathways driving the autoimmunity and tissue fibrosis has yet been fully uncovered. Several novel growth factors and chemokine pathways were also evinced by our analysis, such as the downregulation of IGF1R/IGF1 and PGF/NRP1 and upregulation of HBEGF/CD44, XCL2/XCR1

and CCL21/CCR7 pathways etc. However, the roles of these signaling pathways in mediating autoimmune diseases require further investigation.

Overall, in fibrotic skin, DCs facilitate the most up-regulated receptor/ligand interactions (Fig. 6c, Supplementary Fig. 12a, 97-up/33-down) with other cell types, while CD8⁺ T cells were associated with the most down-regulated interactions (Fig. 6c,

**Fig. 6 Communications between skin resident cell in SSc compare to normal.** (**a**) Example diagram of the communications between dendritic cells, CD4[+], CD8[+] T cells, and fibroblasts through known and predicted SSc pathogenic receptors/ligands which were upregulated in affected skin compare to normal control. (**b**) Circos plot of the ATAC-seq signals around the receptors/ligands in (**a**) in normal versus affected skin from SSc patients. The outermost torus displays the names of the receptors and ligands in each cell type. The middle and inner torus display the ATAC-seq signals at these genes' loci in SSc patients (middle) and healthy controls (inner) respectively. Gene loci that were more accessible in SSc cells were highlighted with shadow. Centre linkers connect the ligands and their receptors between cell types. The width of the linkers represents the length of the corresponding genes. (**c**) Strength of Interaction Alteration (SIA) for each pair of receptor/ligand up-regulated (red) or down-regulated (blue) in an affected state compared to the normal control. Novel receptor/ligand interactions in SSc were highlighted. The bottom rows display the total number of up-regulated and down-regulated receptor/ligand interactions between cell types. Source data are provided as a Source Data file. (**d**–**e**) Normalized ATAC-seq profiles of the normal vs affected CD4 (CD4[+] T cells), CD8 (CD8[+] T cells), DC (dendritic cells) and Fib (fibroblasts) at the *EFNA5* and *EPHA4* (**d**) and the *IL10* and *IL10RA* (**e**) gene loci.

Supplementary Fig. 12c, 43-up/87-down). This analysis highlighted the perturbations of cell-cell communications between DCs and other skin cells in SSc, and provided rich resources of potential drug targets for the treatment of the disease.

## Discussion

Systemic sclerosis is a complex immunogenic and fibrotic disease for which a "driver" cell type and pathway has not yet been identified, leading to a lack of targeted therapies for this disease. Study of the role of the innate immune system in SSc pathogenesis has been particularly limited by mouse models that do not have equivalent skin DC populations to those found in human skin[4]. Study of DCs in SSc has also been limited by the need for ex vivo manipulation of immature blood DC subsets with cytokine maturation factors—a system that does not accurately portray disease-specific pathogenic skin DC subsets. Using ATAC-seq to accurately map open chromatin and transcription factor signaling sites and predict novel interactive pathways on cell subsets within SSc skin samples is a unique and physiological way of looking at the immune and fibrotic landscape in this complex disease.

In this study, we performed ATAC-seq on 8 different skin cell types to survey the cell type-specific regulomes in normal skin as well as affected skin from SSc patients. We observed distinct patterns of DNA accessibility and regulatory networks of transcription factors in different skin resident cells at both healthy and disease states. We also observed that DCs possess the greatest epigenetic differences between the normal, unaffected and affected skin with the most significant enrichment of autoimmune diseases. Analysis of SSc-associated SNPs and skin fibrosis also nominate DCs to be the main epigenetic drivers in the pathogenesis of SSc. Overall, DCs displayed the greatest differential of accessible peaks with relevance to autoimmune disease; the strongest alteration of cell-cell receptor/ligand interactions; the most significant of disease-associated SNP enrichment; the strongest correlation with skin fribrosis; and were also found to highly infiltrate affected skin, suggesting that DCs, which are the primary antigen-presenting cells (APCs) that connect innate and adaptive immunity, may have the greatest impact on skin lesions at an epigenetic level. Although further investigation is still required to fully uncover the mechanisms underlying skin fibrosis, our study provides a better understanding of the functions of DCs, especially cDCs in driving SSc and a rich source of candidates for therapeutic targets to treat the disease.

The present study has several limitations. First, we studied relatively small number of patients, in part because we aimed to isolate multiple primary cell types from lesional and clinically unaffected skin from each patient. As a result, this study is not powered to address potential difference associated with known variables in the disease, such as disease subtypes, autoantibody repertoires, or the stage of progression from early inflammatory to later sclerotic disease. Rather, the features that we identified in

this cross sectional study are likely broad present in SSc, while additional yet to be discovered epigenomic changes drive additional features indicated above. Second, while comprehensive in our initial survey, our cell type-specific analyses were not able to address all relevant cell types in SSc, in particular macrophages that are present in small numbers in normal skin, which preclude a comparison of macrophage epigenomic state in normal vs. SSc skin. Furthermore, cell types that are just now being recognized and not part of the flow-cytometry based prospective isolation were not studied. Thus, much remains to be learned about the complexities of and treatment for SSc. The knowledge generated in this work sets the stage for future efforts to address these outstanding and important questions.

## Methods

**Cell isolation**. Before the skin obtaining, informed consent was obtained from each patient. Ethical approval was obtained from the Stanford Institutional Review Board (IRB) (No.27804). Informed consent was obtained. Biopsies of clinically affected and unaffected skin were obtained from arms and backs of SSc patients respectively. Due to our prior work with macrophage in both normal human skin as well as in inflammatory skin conditions[44,45], we realized that there would be many fewer inflammatory cells (CD163[+] macrophages and CD11c[+]HLADR[+] DCs in particular) in normal human skin compared to scleroderma and so in order to get enough cells for analysis from normal human skin, we utilized discarded abdominoplasty and surgical dog ear tissue (aka large areas of normal skin) to collect enough cells for analysis, whereas we used a 5 mm punch biopsy from scleroderma patients. 5 mm skin punch biopsies were digested overnight in dispase separating the epidermis and dermis, each of which were further digested separately into single cell suspension using standard protocols without ex vivo expansion. Distinct cell types were purified by flow cytometry (FACSAria II Flow Cytometer, Collection: BD FACSDiva, Software Analysis: FlowJo version 10.6.0). Prior work from our group has identified distinct non-overlapping populations of antigen presenting CD45[+]CD11c[hi]HLA-DR[hi]BDCA-1[+] DCs, and phagocytic CD45[+]CD11c[−]CD163[+]FXIIIA[+] macrophages in normal human skin. DCs in inflammatory skin conditions do not express BDCA-1, but do express similar high levels of CD11c and HLADR. In order to maintain comparability across normal and disease states, we used a unified sorting protocol for both normal and inflammatory skin samples. The dermis was sorted into 6 populations: dendritic cells as CD45[+]CD11c[hi]HLA-DR[hi], macrophages as CD45[+]CD11c[−]CD163[+], CD4[+] T cells as CD45[+]CD3[+]CD4[+]CD8[−], CD8[+] T cells as CD45[+]CD3[+]CD4[−]CD8[+], endotheliocytes as CD45[−]CD31[+], and fibroblasts as CD45[−]CD31[−]. The epidermis was sorted into CD45[+]CD1a[+] LCs and CD45[−]CD1a[−] KCs. The samples were sorted in their entirety in order to provide enough material for the ATAC-seq protocol. Post sort purity was >99%. We performed ATAC-seq on each cell type to map the location and accessibility of active DNA regulatory elements genome-wide.

**ATAC-seq library construction and sequencing**. ATAC-seq was performed as described[7]. Briefly, skin cells were sorted using a FACSAria II Flow Cytometer. Samples were lysed in cold lysis buffer (10 mM Tris–HCl (PH 7.4), 10 mM NaCl, 3 mM MgCl$_2$, and 0.1% IGEPAL CA-630) for 3 min on ice to prepare the nuclei. Immediately after cell lysis, nuclei were centrifuged at $500 \times g$ for 10 min using a refrigerated centrifuge and the supernatant was discarded. Nuclei extract were then incubated with the generated Tn5 transposome, 2× TD buffer, and nuclease-free water at 37 °C for 30 min. After DNA purification with the MinElute Kit (Qiagen), PCR were performed to amplify the library for 10–12 cycles distinctively according to a quantitative PCR reaction for optimum cycles. PCR condition was set as, 72 °C for 5 min; 98 °C for 30 s; and thermocycling at 98 °C for 10 s, 63 °C for 30 s and 72 °C for 1 min. When PCR was accomplished, 2 × 50 paired-end sequencing performed on NextSeq 500 (Illumina) to yield, on average, 30 M reads/sample.

**Primary data processing and peak calling**. ATAC-seq raw data was processed using the published ATAC-seq pipeline ATAC-pipe[13]. Sequencing reads were mapped using the "—MappingQC" module in ATAC-pipe. Adapter sequences were trimmed and reads were mapped to Hg19 using Bowtie. PCR duplicates were removed as described[7,13]. Mapped reads were then shifted +4/-5bp depending on the strand of the read, so that the first base of each mapped read represented the Tn5 cleavage position. All mapped reads were then extended to 50 bp centered by the cleavage position. Reads mapped to repeated regions and chromosome M were removed. We used the "—PeakCalling" module in ATAC-pipe with options "--p1 3 --q1 5 --f1 1 -w 50", to call peaks using MACS2[46]. Peaks were then filtered and enriched regions were identified as those with a posterior probability of >0.99. Samples from the same cell type classified under the same clinical condition (normal, unaffected or affected) were grouped for peak calling, and peaks for all categories were then merged together to generate a unique peak list. Numbers of raw read counts mapped to each peak at in each sample were quantified by this module in ATAC-pipe. We then obtained an N × M data matrix where N indicates the number of merged peaks, M indicates the number of samples, and the matrix value $D_{i,j}$ represents the raw read counts fall in peak $i$ ($i = 1$ to N) of sample $j$ ($j = 1$ to M). This data matrix was then normalized by "normalize.quantiles" function of "preprocessCore" package in R, and the normalized matrix was used for downstream analysis.

**Differential analysis**. Peak intensity was defined as $\log_2$ of the normalized read counts. T test and Benjamini–Hochberg multiple test were used to calculated the P value and FDR between any pair of samples.

**Regulome divergence between biopsy versus cultured cells**. We sought to quantify the potential differences in the chromatin landscape of cells directly harvested from fresh skin compared to cells from tissue culture. For fibroblast, cultured samples were represented by ATAC-seq data collected from the human skin fibroblast BJ cell line (GSE81807)[47]. Concomitantly, primary data processing and peak calling was performed on ATAC-seq data of biopsy fibroblasts that from healthy donors and human BJ cell line. We then applied above-mentioned differential analysis to discern the epigenetic differences between fresh versus cultured fibroblasts (P value < 0.05, $|\log_2$ Fold change$| > 2$). Same analysis was performed on keratinocytes from skin biopsy and cultured.

**Cell type-specific peak analysis and functional annotation**. Differential analysis was applied for each cell type compared with all other cell types and cell type-specific peaks were filter with P value < 0.005 (Student T-test), $\log_2$ fold change of mean peak intensity > 2, coefficient of variation of samples in each cell type (inCov) < 0.5, and coefficient of variation of peak intensity among all samples of other cell types (outCov) < 0.5. Bed files of peak lists for each cell type were uploaded to GREAT (version 3.0.0)[16], all gene ontology and relevant functional analysis were performed with options "associating genomic regions with genes 50 kb (basal plus extension model)". Non-repetitive top enriched biological function were manually selected and presented. Cell type-specific peaks and their enriched biological functions for each cell type in healthy controls were identified and summarized in Supplementary Tables 4, 5.

**Construction of cell type-specific transcriptional regulatory networks**. All known motifs of vertebrate transcription factors were obtained from the Jaspear database[19]. We identified their genome-wide putative occupants using "findMotifsGenome.pl" script in HOMER, and converted the results into a 0/1 matrix, where each row is a peak and each column is a motif. The matrix value $D_{p,m}$ represents whether motif m was identified in peak p (0 means False, 1 means True). We then input this peak by motif matrix and the normalized peak read count matrix into Genomica (https://genomica.weizmann.ac.il/), and applied the "ModuleMap" algorithm to calculate the degree of enrichment for every motif in every sample, and thus constructed cell type-specific regulatory networks.

**Disease-associated SNP enrichment analysis**. To evaluate the enrichments of the disease-associated variants in cell type-specific open chromatin and regulatory regions, we used all relevant SNVs from GWAS experiments in the GRASP database[15,48] (GRASP 2.0.0.0: https://grasp.nhlbi.nih.gov/Updates.aspx), GWAS database (GWAS catalog: https://www.ebi.ac.uk/gwas) and recently published SSc associated GWAS study[49]. 9,026,521 genotype-phenotype results and 188,362 unique phenotypes were collected, with P value < 0.05. To analyze the enrichment of each disease with each ATAC-seq sample, we first obtained the genomic positions of the disease-associated SNP sets from the GRASP database, filtered them at different levels of P-value thresholds from 0.05, $10^{-3}$, $10^{-5}$, $10^{-6}$ to $10^{-8}$, and got sets of disease-associated SNPs at different degrees of significance. Next, we used the "-intersect" option in bedtools and overlapped these filtered SNP sets with all ATAC-seq peaks, and obtained a disease by peak matrix. Diseases whose associated SNPs were less than 20 were discarded to avoid statistical errors. Next, we calculated the "deviation score" (defined in Jason D. Buenrostro et al.[50]) for every remaining disease for each sample, and then average "deviation score" is used to quantify the enrichment of disease-associated SNP in each cell types.

**Cell type-specific peak in SSc**. For each cell type, pairwise comparison between groups of samples classified into different clinical states were performed. Lists of differential peaks were obtained and categorized into 6 groups: those enriched in normal only, unaffected only, affected only, normal and unaffected, unaffected and affected, and normal and affected samples. Significant peaks were defined as $|\log_2$Fold change$|$ between groups of samples >0.8, and ranges of peak intensities within each group of samples <1.5.

**Correlation analysis of cell-type signature score and gene expression**. Signature genes for each cell type was obtained by anchoring cell-type specific peaks (in both control and SSc) to their closest genes through "annotationPeaks.pl" in HOMER. Modified Rodnan skin score (mRSS) and bulk cell gene expression microarray data of lesion skin biopsies of the SSc patients in the treatment of mycophenolate mofetil (MMF) were obtained from GEO database (GSE76886)[26]. All of the 105 arm samples from 30 MMF-treated patients were included in the correlation analysis (Supplementary Table 10). Microarray gene expression matrix was filtered and normalized according to published methods[26]. To remove the impact of MMF treatment, we first obtained the MMF response genes (list in Supplementary Table 9) through the differential analysis between the skin samples of 12/24 months treatment and baseline ($|\log_2$ Fold change$| > 2$, P value < 0.005). Then, we removed MMF response genes from the signature genes of each cell type. Cell type signature score was defined as the average normalized gene expression of the signature genes (with MMF response genes removed) from the microarray data obtained from SSc patients' affected skin. Pearson correlation of the signature score and mRSS for each cell type were shown. In Fig. 2e, for each patient during the treatment, we identified the time points when the mRSS score is the lowest (time point Low) and highest (time point High), and then calculated the corresponding signature scores for each cell type from the microarray profiles at time point Low and time point High, respectively. The time points Low and High can be different for different patients. In this analysis, 13 patients whose highest mRSS - lowest mRSS > 5 were included (patients' ID 03, 04, 05, 06, 10, 16, 17, 21, 30, 33, 37, 42, 45).

**TF Foot-printing analysis**. TF foot-printing analysis was processed using the published ATAC-seq pipeline ATAC-pipe[11]. Briefly, the HOMER "scanMotifGenomeWide.pl" script was used to identify the genome-wide motif occupancy sites using the default settings. We normalized the ATAC-seq reads in each category of samples (normal, affected and unaffected cells) by randomly selecting 500 M reads from each group. The average coverage of each motif around a 100 bp genomic region centered by the motif sites was calculated, consistent with previously reported methods[51].

**Analysis of resident skin cell communication through receptor-ligand interactions**. We first defined a connection between cell type A and B if the chromatin around a receptor in cell type A and its ligand in cell type B were both accessible. If their accessibilities were both significantly higher/lower in SSc compared to normal controls, then this connection was considered as altered through the corresponding pair of receptor/ligand partners. Known intercellular receptor/ligand partners were obtained from the Ligand-Receptor Partners (DLRP) database[40] and the CellPhoneDB[41]. Each peak was imputed to its nearest gene by HOMER using the "annotationPeaks.pl" script, and the expression levels of every gene were measured by summing the intensities of all peaks that were imputed to this gene. To quantify the alteration of the strength of receptor/ligand in the disease state compared to normal, we defined an Alteration Score (AS) for each gene as the fold change of the predicted expression in affected cells versus normal control. We next calculated the AS for all pairs of receptors/ligands. We then defined the Strength of Interaction Alteration (SIA) between cell types as follows Eq. (1):

$$SIA = \begin{cases} 0 (if \left| AS_{receptor} \right| < 1 \ or \ \left| AS_{ligand} \right| < 1) \\ AS_{receptor} + AS_{ligand} (else) \end{cases} \quad (1)$$

The strength of alteration greater than 7 or less than -7 were retained for downstream analysis. A positive strength of alteration indicates an up-regulated interaction between the cells, and a negative strength of alteration suggests a down-regulated interaction. The "circular" package (version 0.4.11) in R was used to draw the circus plots.

**Reporting Summary**. Further information on research design is available in the Nature Research Reporting Summary linked to this article.

## Data availability

The raw data and pre-processed ATAC-seq data matrix of this study have been deposited in Gene Expression Omnibus(GEO) with the primary accession code GSE99702. We also used other published datasets in GEO including (1). ATAC-seq data collected from the human skin fibroblast BJ cell line, the accession number is GSE81807[47]; (2). Modified Rodnan skin score (mRSS) and bulk skin cell gene expression data of lesion skin biopsies of the SSc patients in treatment of mycophenolate mofetil (MMF), the accession number is GSE76886[26]; (3). Bulk skin cell gene expression data of normal and lesion skin, the

accession numbers are GSE130955[33], GSE58095[34]. Other data used in our paper: (1). hg19 reference genome and annotation were downloaded from UCSC [http://hgdownload.cse.ucsc.edu/goldenPath/hg19] and refseq [https://www.ncbi.nlm.nih.gov/projects/genome/guide/human]; (2). All TF motifs were obtained from HOMER (Motif Analysis tools) homepage [http://homer.ucsd.edu/homer/motif/]; (3). All interactions were downloaded from Ligand-Receptor Partners(DLRP) database [https://www.allacronyms.com/DLRP/Database_of_Ligand-Receptor_Partners] and the CellPhoneDB [https://www.cellphonedb.org/explore-sc-rna-seq]; (4). All published disease associated-SNPs were obtained from GRASP 2.0.0.0 [https://grasp.nhlbi.nih.gov/Updates.aspx] and GWAS database [GWAS catalog:https://www.ebi.ac.uk/gwas]. Source data are provided with this paper.

## Code availability

Processed code is available in https://github.com/QuKunLab/SSc-ATAC-seq.

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

## Acknowledgments

We thank the members of Chang and Qu Labs for their discussions, Grant Ognibene and Illisha Rajasansi for coordinating and collecting patients' samples, and the patients involved in this study for their participation. This work was supported by the National

Key R&D Program of China 2017YFA0102900 (to K.Q.), the National Natural Science Foundation of China grant (91940306, 81788101, 31970858, 31771428 and 91640113 to K.Q., 31700796 to C.G. and 81871479 to J.L.), the Scleroderma Research Foundation (H.Y.C.), NIH grant P50-HG007735 (H.Y.C.), the Fundamental Research Funds for the Central Universities (WK2070000158, YD2070002019, WK9110000141 to K.Q.) and the Dermatology Foundation (L.Z.). We thank the USTC supercomputing center and the School of Life Science Bioinformatics Center for providing supercomputing resources for this project. We thank the CAS interdisciplinary innovation team for helpful discussion. H.Y.C. is an Investigator of the Howard Hughes Medical Institute.

## Author contributions

H.Y.C., K.Q., and L.Z. conceived the project. L.Z., A.T.S., M.L. performed all cell sorting. L.Z., R.L., and K.T. performed ATAC-seq library generation. Q.L. performed all data analysis with assistant from W.Z., K.L., J.G., C.G., and J.L. G.K. and O.D. performed protein staining experiments. D.F. and L.C. provided clinical assessments of the disease. Q.L., K.Q., H.Y.C., L.Z., and M.L. wrote the manuscript with inputs from all authors.

## Competing interests

H.Y.C. is affiliated with Accent Therapeutics (co-founder and advisor), Boundless Bio (co-founder and advisor), 10x Genomics (advisor), Arsenal Biosciences (advisor), and Spring Discovery (advisor).
