## [Peer Review File · Nature Communications]

REVIEWER COMMENTS

Reviewer #2 (Systemic sclerosis, clinical and models of pathogenesis)(Remarks to the Author):

This is a very well-presented paper that describes use of ATACseq to profile cells isolated from SSc or control skin. A small cohort is examined and SSc associated differences identified in relevant cell types. The most distinct differences are seen for skin dendritic cells. This fits with the autoimmune nature of SSc and gives additional insight into the relevance of epigenetic mechanism that may be cell type specific and relevant to pathogenesis.

Specific points

1. The SSc cohort is very small and for a diverse disease it is a limitation that shared differences may not include those relevant to a particular age or subset.
2. It would be interesting to look at ANA subtypes especially as these could relate to antigen driven processes relevant to skin dendritic cells.
3. The absence of functional studies is a key limitation. The work is essentially descriptive and supports the role of cell specific chromatin changes relevant to gene regulation being implicated in systemic sclerosis pathogenesis. However, this is not in itself novel. The study confirms candidates that have emerged from other work and methodologies. It would be substantially strengthened by additional functional and confirmatory experiments using relevant cells or in vivo animal systems.
4. Many of the candidate pathways are not new and have been implicated in many other studies. This should be discussed, and the additional or more novel aspects of the present work could be explained in greater detail. The discussion section is very brief and could be expanded.
5. It would be helpful to have better description and characterisation of the endotheliocyte population.

Reviewer #3 (Systemic sclerosis, genomics)(Remarks to the Author):

Liu et al. applied ATAC-seq to examine the differences of 8 immune cell types between healthy skin, affected SSc skin and unaffected SSc skin. By identifying differential peaks, TF enrichment and correlating with MRSS, the authors identified that dendritic cells could be essential in SSc pathogenesis. The authors validated their findings in vivo by comparing ZBTB46 expression between SSc skin and normal skin. This study adds to a growing body evidence that implicates myeloid cells of the innate immune system as key drivers of SSc, some implicating DCs and others macrophages. Because of this literature suggesting different cell types may be involved, it is important that the authors clearly state the limitations of this study, which is that it doesn't necessarily exclude other cell types not well studied here from being important in SSc pathogenesis. Overall, this is an important and well done study, but I have a number of points that should be addressed.

Major comments:

1. The authors initially isolated cells from normal skin. The number of DCs and macrophages in the normal tissue should be exceptionally small and I would expect increased numbers of cells in the SSc patient samples. How did the number of cells isolated from the normal tissue compare to the numbers of cells isolated from the SSc tissues? Did these differ by diffuse and limited SSc?
2. The sorting and gating approach shown in supplemental figure 1 shows clear cell populations for 7 cell types, but the panel and gating for macrophages shows no specific population that is selected and the authors appear to have selected cells at the edge of a population. Based on the data presented, I am skeptical that a true macrophage population was isolated. How did the authors confirm they had isolated an population of macrophages? Can the authors also show the flow sorting for the SSc

samples?

3. The clustering of cell types shown in figure 1C has the DC and LC populations grouping together as I would expect, but the macrophages cluster with the endothelial cells which is not what I would expect. The macrophage population is from a single isolate in contrast to at least duplicate or triplicate for other cells. Given this is only from a single sample, it is very hard to draw any conclusions regarding this cell population. Can the authors explain why these macrophages cluster with the endothelial cells? I am surprised that this is also true in the PCA plot of the SSc samples.

4. It will be helpful to have a supplemental table summarizing the immune cells analyzed, abbreviations, number of replicates, number of reads and number of peaks for normal skin, involved and un-involved skins in SSc.

5. In figure 2A, the authors show enriched peaks for nearly all of the cells but there doesn't appear to be a specific peak for the macrophage population. Enriched GO terms are also not presented. Why is this the case?

6. Please provide a supplemental table listing all significant peaks for each cell type in both control and SSc. Please also provide a supplemental table listing the enriched GO terms for each.

7. Please provide the list of the SSc SNPs that were interrogated for this study. It would be helpful to have the list for each of the sets that were interrogated.

8. The correlation between cell signature scores and MRSS makes use of a publicly available DNA microarray dataset but the details are only in the methods, and this is confusing. Please put in a sentence describing that the analysis is using publicly available data and the skin scores from that dataset.

9. In Fig. 2D, the publicly available dataset used is an analysis of MMF treatment over time and MMF treatment impacts the immune signatures, particularly those associated with T cells and macrophages. The authors need to take treatment into account to ensure the lack of correlation with other cell types is not a result of an impact on their gene expression signatures by MMF treatment.

10. I am also concerned with the statement in the methods that "only data of MMF responder with improved MRSS during treatment were used for subsequent analysis". The rationale for using only these patient in Figure 2 are unclear and it would result in biased set of patients. The low vs high analysis shown in Figure 2e states highest and lowest values for each patient but there is no information on the time point used.

11. Please provide a supplemental table of the signature genes of immune cells used in Fig. 2D.

12. Is the ZBTB46 gene also enriched in the DC cell specific differential peaks? Given the heterogeneity observed in the staining (Fig 3L), is there a trend for ZBTB46 expression in other SSc skin databases? Did you examine MafB, which is macrophage marker?

13. Do the authors have some explanation on the overlap of unaffected and affected skins in Fig. 3A-D? DC cells seem have a much more significant enrichment in unaffected SSc skin. This seems counter to the conclusions drawn from this figure.

14. In figure 3, "Interestingly, we noticed: (1) 28.5% of the differential peaks in fibroblast consist of peaks which represent a healthy cell state (cluster 2), but only 1.9% and 5.8% of these peaks were

observed in T cells and DCs, respectively. This result was expected since fibrosis is the main distinction between clinically affected and unaffected skin." This is hard to understand. First, do the authors mean that peaks for each cluster across immune cells are the same? From the plots, it doesn't look it that. Second, there are a large fraction of peaks are overlapped in unaffected skin and normal skin in cluster 2 for Fib. Second, what genes are enriched in the cluster 2 of Fib? Can the 28.5% overlap allow you to draw the conclusion that "This result was expected since fibrosis is the main distinction between clinically affected and unaffected skin."

15. I am concerned that the SSc samples represent a very small number of patients and that the authors have lumped samples from patients with limited cutaneous SSc and diffuse cutaneous SSc together. There is significant heterogeneity within these two clinical groups. To this end, the TF motifs show in Figure 3I for DCs seem to be enriched in one patient or another, but in only one case (NFkB) does it appear to be consistent across patients. This heterogeneity in the results needs to be explained within the context of the clinical and molecular subtypes. Are the DCs similarly enriched in both diffuse and limited SSc patient samples? Is there difference between the DCs of these two patient groups? In addition, figure 3L which quantifies the levels of ZBTB46 in SSc patients shows 5 patients with levels very similar to controls and 3 patients with high expression. This suggest significant variability of the DC population across SSc patients (see comment 12 above).

16. The authors state on page 9, "in this way, we can systematically delineate all potential communications between T cells, DCs and Fibs in SSc". This seems an over reach given the small number of SSc samples, and the variation among patients that are in this study. The number of patients examined here is vanishingly small. This statement should more clearly indicate this is a hypothesis generating exercise on a small number of patients.

17. The conclusions of the paper need to clearly indicate the limitations of the study, in particular that other cell types, such as macrophages or cell types not isolated by their initial flow procedure (maybe some that are only now being discovered by single cell methods), can't be excluded based on this study.

Minor comments:

1. Suggest, changing the title of Fig. 1 to "Landscape of DNA accessibility in 8 cell types from normal skin in vivo".
2. How did you define the highest and lowest MRSS in Figure 2E. Please add this information to the figure legend for figure 2.
3. Highlight the gene loci with more accessible in SSc in Fig. 4b.
4. Fig. 4b is hard to understand. Need a better interpretation in the Results part.
5. In the methods, the publicly available dataset is mentioned as RNA-seq, when the data are DNA microarray based.

Michael Whitfield

Responses to reviewers

We thank the reviewers for their positive assessment and thoughtful suggestions that have improved this work. Point-by-point responses are as follows.

Reviewer #2 (Systemic sclerosis, clinical and models of pathogenesis) (Remarks to the Author):

This is a very well-presented paper that describes use of ATACseq to profile cells isolated from SSc or control skin. A small cohort is examined and SSc associated differences identified in relevant cell types. The most distinct differences are seen for skin dendritic cells. This fits with the autoimmune nature of SSc and gives additional insight into the relevance of epigenetic mechanism that may be cell type specific and relevant to pathogenesis.

Response: Thank you for the positive feedback and constructive suggestions to improve our manuscript.

Specific points

1. The SSc cohort is very small and for a diverse disease it is a limitation that shared differences may not include those relevant to a particular stage or subset.

Response: We agree that the patient cohort size is small in our study. Our goal was to identify the epigenomic profiles of multiple distinct primary cell types from SSc lesions, and thus each biopsy sample requires extensive cell sorting and purification to transform into multiple downstream data sets. This design is not suited for cross-sectional study of large number of patients. Our study employed a similar sized cohort as other molecular studies of human SSc tissue recently published in top journals, including Shin, J.Y. et al. *Science Translational Medicine*, 2019 (n=3 SSc patients by RNA-seq of dermal fibroblast)¹; Wohlfahrt, T. et al. *Nature*, 2019 (n=9 SSc patients by RNA-seq of dermal fibroblast)²; Maehara, T. et al. *The Journal of Clinical Investigation*, 2020 (n=3 SSc patients by RNA-seq of peripheral CD4⁺ naïve and cytotoxicity T cell)³. Our study includes 28 samples from 7 SSc patients, where we produce the first genome-wide chromatin maps (in contrast to exome analysis focused on only ~2% of human genome) for multiple skin cell types rather than bulk skin cells. Our study provides a valuable resource that systematically described the chromatin profile differences of multiple cell types among normal, clinically affected and unaffected skin in SSc patients. We observed clear chromatin signatures for enriched DCs and altered T cell patterns in SSc lesions, which are likely to be shared in many subtypes of SSc. The reviewer is correct that we are not powered to detect epigenomic changes in particular stages or subtypes of SSc, and our study sets the stage for future studies to address these questions. We have added a paragraph in **Discussion** to highlight this point.

2. It would be interesting to look at ANA subtypes especially as these could relate to antigen driven processes relevant to skin dendritic cells.

Response: We thank the reviewer for this interesting suggestion. We have provided detailed ANA subtype spectrum for all SSc patients in **Supplementary Table 2**. Affected dendritic cells were extracted from three SSc patients F28, M25, F1205, and their ANA were (ANA⁺, RNAPolII⁻, SCL70⁺, anti-centromere⁻), (ANA⁺, RNAPolIII⁺, Scl70⁻, anti-centromere⁻), (ANA⁺, RNAPolIII⁻, SCL70⁻, anti-centromere⁺) respectively. Due to the limitation of the sample size, we are not sufficiently powered to evaluate a connection between ANA subtypes and the abnormal antigen procession of skin dendritic cells.

3. The absence of functional studies is a key limitation. The work is essentially descriptive and supports the role of cell specific chromatin changes relevant to gene regulation being implicated in systemic sclerosis pathogenesis. However, this is not in itself novel. The study confirms candidates that have emerged from other work and methodologies. It would be substantially strengthened by additional functional and confirmatory experiments using relevant cells or in vivo animal systems.

Response: We believe that the value of our work lies in uncovering the chromatin state of primary cell types in SSc lesions from patients. These data complement insights from mouse models or tissue culture cells, which often deviate from the human disease. This study provided three types of validation. First, an important mechanistic insight from this work is the demonstration that human inherited variants (SNPs) associated with SSc propensity are most enriched in active DNA regulatory elements in lesional dendritic cells (DC). Second, we provide an independent cohort of patient samples to validate increased DC in SSc lesions as analyzed by immunohistochemistry (IHC). Third, we described a computational model to explore the changed receptor-ligand interactions at affected SSc skin. As the reviewer mentioned, there are many well-studied receptor-ligand signal pathways appear in our results, such as the NOTCH and TGF β signal pathways, which fully supported the reliability of our data and analysis. Meanwhile, we also found several novel signaling pathways such as the HBEGF-CD44, XCL2-XCR1, CCL21-CCR7 interactions, which have not been reported in the published studies. Our analysis illuminated that in fibrotic skin of SSc patients, DC play a central role in communication between skin resident cells, which has also not been reported. We recognize the reviewer's point, and have toned down the conclusions that our work provides a resource for hypothesis nomination, and should be tested by further functional studies in the future. We also provided a paragraph to discuss the limitations of our study in **Discussion**.

4. Many of the candidate pathways are not new and have been implicated in many other studies. This should be discussed, and the additional or more novel aspects of the present work could be explained in greater detail. The discussion section is very brief and could be expanded.

Response: We thank the reviewer for these suggestions which are very helpful to improve our manuscript. We have performed literature search on the predicted pathways and highlighted the novel receptor/ligand interactions in the revised **Fig. 4C**. For example, signaling pathways such as IGF1R/IGF1 and PGF/NRP1; and HBEGF-CD44, XCL2-XCR1 and CCL21-CCR7, which have not been reported to be associated with SSc from other published studies, were found significantly down/up-regulated in SSc in our study. However, since our study contains relatively small number of patients and lacks additional experimental confirmations, the pathogenic mechanisms for these novel signaling pathways in mediating autoimmune diseases require further investigation. We have addressed the limitations of this study in the revised **Discussion**.

5. It would be helpful to have better description and characterization of the endotheliocyte population.

Response: We thank the reviewer for this constructive suggestions. We have performed a correlation analysis of the signature score of each cell type versus the degrees of fibrosis of the skin from SSc patients, measured by the modified Rodnan skin scores (mRSS), we found that signature scores of only DC and T cell rather than endotheliocyte (EC) were significantly positive correlated with mRSS (**Fig. 2d-e**). From the SNPs enrichment analysis, we found a significant enrichment of SSc risk SNPs in chromatin accessible regions of DC rather than EC (**Supplementary Fig. 3**). Therefore, from our data analysis, the epigenome of the skin EC has little effect on the aggravation of the SSc. Furthermore, we also performed differential analysis between normal, unaffected and affected EC by the same method used in **Fig. 3a-d** without statistical power since there was only 1 sample in unaffected and affected skin were obtained. We obtained a total of 25934 differential peaks ($|\log_2 \text{ Fold change}| > 2$), however the up-regulated peaks in affected EC were not enriched to any autoimmune fibrosis relevant GO terms (**Graph 1**). We include this analysis in the revised manuscript as a supplementary figure (**Supplementary Fig. 4d**).

Graph 1. Cell types-specific regulome divergence in normal, unaffected and affected skins. (a). Heatmaps of the normalized ATAC-seq intensities (Z-scores) of peaks enriched in normal, unaffected and affected EC. Cluster 1-6 represent the peak groups enriched in normal only, normal and unaffected, unaffected only, unaffected and affected, affected only, and normal and affected cells respectively. Each row is a peak and each column is a sample. (b). Bar plot showing the disease annotation of peaks enriched in cluster 5.

Reviewer #3 (Systemic sclerosis, genomics) (Remarks to the Author):

Liu et al. applied ATAC-seq to examine the differences of 8 immune cell types between healthy skin, affected SSc skin and unaffected SSc skin. By identifying differential peaks, TF enrichment and correlating with MRSS, the authors identified that dendritic cells could be essential in SSc pathogenesis. The authors validated their findings in vivo by comparing ZBTB46 expression between SSc skin and normal skin. This study adds to a growing body evidence that implicates myeloid cells of the innate immune system as key drivers of SSc, some implicating DCs and others macrophages. Because of this literature suggesting different cell types may be involved, it is important that the authors clearly state the limitations of this study, which is that it doesn't necessarily exclude other cell types not well studied here from being important in SSc pathogenesis. Overall, this is an important and well done study, but I have a number of points that should be addressed.

Response: We thank the reviewer for their positive assessment that our work is “an important and well done study” to the pathogenesis of myeloid cells to SSc. We appreciate the reviewer's constructive suggestions to improve our manuscript.

Major comments:

1. The authors initially isolated cells from normal skin. The number of DCs and macrophages in the normal tissue should be exceptionally small and I would expect increased numbers of cells in the SSc patient samples. How did the number of cells isolated from the normal tissue compare to the numbers of cells isolated from the SSc tissues? Did these differ by diffuse and limited SSc?

Response: DCs are indeed increased in number in SSc skin lesions, which we show using ZBTB46 immunohistochemistry (**Fig. 3k-l**). In addition, DCs from SSc lesion show altered chromatin accessibility, including at binding sites for transcription factors NF- κ B, STAT1, and noncoding DNA elements associated with SSc by genome-wide association studies (**Fig. 3i-j**). The reviewer is absolutely correct that macrophages are rare in normal skin, and we could not consistently sort and isolate enough macrophages to compare to SSc skin to make quantitative conclusions. In fact, we were able to isolate

macrophages from normal lesion with sufficient number to generate ATAC-seq data that passed our quality control only a single time, and once each from diffuse and limited SSC lesion (**Supplementary Tables 1-3**). We included the macrophage data because this was part of our study intent and for interest to the field, but the number is obviously not sufficient to support any definitive conclusion. If the reviewer feels that this information is distracting, we are open to removing the macrophage data.

2. The sorting and gating approach shown in supplemental figure 1 shows clear cell populations for 7 cell types, but the panel and gating for macrophages shows no specific population that is selected and the authors appear to have selected cells at the edge of a population. Based on the data presented, I am skeptical that a true macrophage population was isolated. How did the authors confirm they had isolated a population of macrophages? Can the authors also show the flow sorting for the SSC samples?

Response: As indicated above in point 1, we were able to isolate macrophages from normal and diffuse and limited SSC lesion with sufficient number to generate ATAC-seq data that passed our quality control only a single time each. Nonetheless, we followed the CD163⁺ macrophage sorting strategy from skin, which was previously used to characterize macrophage population in psoriatic skin (Zaba et al., JCI, 2007; Duculan et al., JID, 2010)^{6,7}. In SSC skin where number of macrophages is increased, we can confidently sort out this population (**Graph 2**).

Graph 2. Gating strategy used to define cell types of normal (a) and SSC (b) samples from FACS.

3. The clustering of cell types shown in figure 1C has the DC and LC populations grouping together as I would expect, but the macrophages cluster with the endothelial cells which is not what I would expect. The macrophage population is from a single isolate in contrast to at least duplicate or triplicate for other cells. Given this is only from a single sample, it is very hard to draw any conclusions regarding this cell population. Can the authors explain why these macrophages cluster with the endothelial cells? I am surprised that this is also true in the PCA plot of the SSc samples.

Response: Since dermal macrophages and dendritic cells were both differentiated from monocytes, and our normalized ATAC-seq profiles showed that peaks around several marker genes of myeloid cells, such as *ITGAX*(CD11C), *CD80*, *CD68*, *HLA-DRA*, *TLR4* were more accessible in macrophages and myeloid dendritic cells than other cell types (**Graph 3a**). This indicated that our ATAC-seq data of macrophage is reliable. However, the correlation analysis across different cell types of healthy controls and SSc patients both showed strong correlation between macrophages and CD31⁺ endothelial cells (**Graph 3b**). Furthermore, signature peaks of macrophages and endothelial cells were both enriched the biological functions about angiogenesis (**Graph 3c-d**). Macrophages are very plastic cells, and one aspect of its heterogeneity is the tissue specialization of resident macrophages⁸. The dermal macrophages have been reported involved in angiogenesis through expression of vascular growth factor⁹. Our results further suggested that the epigenetic regulome of macrophage residing in the dermal layer are very different from that of other myeloid cells but similar to that of endothelial cells.

Graph 3. The chromatin accessible pattern of macrophage and other skin cell types. (a). Normalized ATAC-seq profiles around *ITGAX*, *CD80*, *CD68*, *HLA-DRA*, *TLR4*.

(b). Unsupervised hierarchical clustering of the Pearson correlations between all the samples. ATAC-seq signals were obtained from distal elements (distance to promoter >1 kb). Left panel showing the correlation across samples of all cell types shared by healthy controls and SSc patients. Upper right and lower right panel showing the correlation across all normal and SSc affected samples respectively. (c). Heatmap of the normalized ATAC-seq intensities of cell type-specific peaks from healthy control and SSc patients. Each row is a peak, and each column is a sample, with color coded cell types (top panel). Clusters shown in the sidebar represent cell type-specific peaks of T cells (C1), dendritic cells/Langerhans cells(C2), macrophages/endotheliocytes (C3), fibroblasts (C4) and keratinocyte (C5) respectively. Functional marker genes in each cluster were shown on the right. (d). Top enriched GO terms of peaks in each cluster with *P* values obtained from GREAT.

4. It will be helpful to have a supplemental table summarizing the immune cells analyzed, abbreviations, number of replicates, number of reads and number of peaks for normal skin, involved and un-involved skins in SSc.

Response: We thank the reviewer for these suggestion. We have summarized this information in **Supplementary Tables 1-3**.

5. In figure 2A, the authors show enriched peaks for nearly all of the cells but there doesn't appear to be a specific peak for the macrophage population. Enriched GO terms are also not presented. Why is this the case?

Response: We have only 1 sample of macrophage in normal control, and thereby unable to screen out the macrophage-specific signature peaks with statistical power. We then did not include macrophage in the previous **Fig. 2a**. We state the limitations of our study in the revised **Discussion**. When we screened for macrophages-specific peaks by combining samples from both the normal donors and SSc patients, we still observe similar clustering patterns and enriched GO terms on macrophages (**Graph 3c-d**, see response to comment 3 above).

6. Please provide a supplemental table listing all significant peaks for each cell type in both control and SSc. Please also provide a supplemental table listing the enriched GO terms for each.

Response: We thank the reviewer for this suggestion. We summarized all the signature peaks and their GO terms for each cell type in both control and SSc in **Supplementary Tables 4, 5**.

7. Please provide the list of the SSc SNPs that were interrogated for this study. It would be helpful to have the list for each of the sets that were interrogated.

Response: We have summarized all SSc susceptibility loci that were interrogated for this study in **Supplementary Table 7**.

8. The correlation between cell signature scores and MRSS makes use of a publicly available DNA microarray dataset but the details are only in the methods, and this is confusing. Please put in a sentence describing that the analysis is using publicly available data and the skin scores from that dataset.

Response: We have added one sentence describing the analysis in the revised main text as below:

*“we downloaded the published microarray gene expression data of SSc affected skin (a total of 105 arm samples obtained from 30 patients) at 3-4 time points along the treatment of mycophenolate mofetil (MMF)¹⁰ and performed a correlation analysis of the average expressions of cell type specific genes (**Supplementary Table 4**) versus the degrees of fibrosis of the skin from SSc patients, measured by the modified Rodnan skin scores (mRSS)¹¹. To remove the impact of MMF treatment on the correlation analysis, genes response to the MMF treatment (**Supplementary Table 9**) were removed from the input gene list before the correlation analysis was performed (see **Methods**).”*

9. In Fig. 2D, the publicly available dataset used is an analysis of MMF treatment over time and MMF treatment impacts the immune signatures, particularly those associated with T cells and macrophages. The authors need to take treatment into account to ensure the lack of correlation with other cell types is not a result of an impact on their gene expression signatures by MMF treatment.

Response: To remove the impact of MMF treatment on the correlation analysis, we first obtained a list of MMF response genes (**Supplementary Table 9**) through differential analysis of the gene expression profiles from the baseline control samples and samples after 12 and 24 months of treatment with MMF. We then excluded the MMF response genes from the previous cell type-specific signature genes (**Graph 4a**), and re-calculated the cell type signature score for the 9 MMF improvers in a same way as before. We found that the revised DC signature score was still the most significantly correlated scores to mRSS (**Graph 4b**). In the revised manuscript, we included all of the 105 arm samples from 30 MMF-treated patients (including the 38 samples from 9 responders as previous submission), and re-performed the correlation analysis in a same way with MMF response genes excluded, and found that the average expressions of DC signature genes were still most significantly positively correlated with mRSS among the six cell types examined (P value= 1.5×10^{-5} , $R=0.41$, revised **Fig. 2d**, **Supplementary Table 10**). These results suggest that our previous conclusion on the relevance of DC to disease pathology was not a result of an impact of MMF treatment.

Graph 4. The correlation between mRSS and signature score of each cell type. (a). Venn Diagram showing the overlap of the MMF response genes and signature genes of each cell type. **(b).** Scatter plot showing the correlation between the signature score of each cell type and mRSS, shaded area represent the 95% confidence interval.

10. I am also concerned with the statement in the methods that “only data of MMF responder with improved MRSS during treatment were used for subsequent analysis”. The rationale for using only these patient in Figure 2 are unclear and it would result in biased set of patients. The low vs high analysis shown in Figure 2e states highest and lowest values for each patient but there is no information on the time point used.

Response: We thank the reviewer’s question and have clarified the patients’ selection in the revised **Methods**. Briefly, in addition to the 38 samples from 9 responders included in the previous submission, we included all of the 105 arm samples from 30 MMF-treated patients in the revised manuscript¹⁰ (**Supplementary Table 10**), and re-performed the correlation analysis in a same way with MMF response genes excluded. We found that the average expressions of DC signature genes were still most significantly positively correlated with mRSS among the six cell types examined (P value= 1.5×10^{-5} , $R=0.41$, revised **Fig. 2d**). These results suggest that our previous conclusion on the relevance of DC to disease pathology was not a result of an impact of MMF treatment nor sample selection. In terms of the low and high values in **Fig. 2e**, we first identified the time points when the mRSS score is the lowest (time point Low) and highest (time point High) for each patient, and then calculated the corresponding signature scores for each cell type from the microarray profiles at time point Low and time point High, respectively for comparison. The time points Low and High can be different for different patients. In this analysis, 13 patients whose highest mRSS - lowest mRSS > 5 were included (patients’ ID 03, 04, 05, 06, 10, 16, 17, 21, 30, 33, 37, 42,45). We have summarized the time point information and cell type-specific signature scores for each patient in the revised **Supplementary Table 10**.

11. Please provide a supplemental table of the signature genes of immune cells used in Fig. 2D.

Response: We summarized the signature genes of each cell type in **Supplementary Table 4**.

12. Is the ZBTB46 gene also enriched in the DC cell specific differential peaks? Given the heterogeneity observed in the staining (Fig 3L), is there a trend for ZBTB46 expression in other SSc skin databases? Did you examine MafB, which is macrophage marker?

Response: Gene *ZBTB46* has been reported as a specific marker gene for myeloid dendritic cells¹², and our ATAC-seq data does indicate that *ZBTB46* is enriched in DC specific differential peaks (**Supplementary Fig. 1f**). We have also seen multiple peaks around the promoter and enhancer regions of *ZBTB46* were specifically more accessible in DC (**Graph 5a, left panel**). Besides, data from multiple studies have shown that the gene expression of *ZBTB46* were significantly higher in SSc skin versus normal skin (**Graph 5b**)^{10,13,14}. These results further support our observation of ZBTB46 staining experiment in **Fig. 3I**. The transcription factor MafB is important to the differentiation of macrophage, and was used as a marker gene to distinguish DC and macrophage¹⁵⁻¹⁷. However, MafB was found expressed in many cell types¹⁵, and we also observed that the promoter region of *MAFB* was highly accessible in multiple cell types in skin (**Graph 5a**). Thereby, we suspected that *MAFB* was not a cell-type specific marker to distinguish macrophage from other cell types in skin. In addition, published data sets indicated that there was no significant difference in terms of the expression of *MAFB* in SSc skin versus normal skin (**Graph 5c**).

Graph 5. ZBTB46 is a marker of myeloid dendritic cells. (a). Normalized ATAC-seq signals at *ZBTB46* (left) and *MAFB* (right) loci. (b-c). The gene expression of *ZBTB46* (b) and *MAFB* (c) in normal and SSc skin. Three panels represent datasets from three different published studies (Mann-Whitney U test, boxplots: 25%, 50%, and 75% quantiles).

13. Do the authors have some explanation on the overlap of unaffected and affected skins in Fig. 3A-D? DC cells seem have a much more significant enrichment in unaffected SSc skin. This seems counter to the conclusions drawn from this figure.

Response: Whitfield et al. first reported that that clinically unaffected SSc skin has similar transcriptional changes as affected SSc skin compared to healthy skin¹⁸. However, the cell type(s) responsible for this phenomenon was not clear. We designed our cell-type specific analyses to interrogate both clinically affected and unaffected skin in SSc patients to potentially answer this question. When we performed a pair-wise comparison of ATAC-seq profiles between clinically affected vs. unaffected vs. normal skin biopsies on T cells, DCs and fibroblasts, we noticed that DCs exhibited the most epigenomic diverges between healthy and disease states, with a total of 15869 significant peaks compare with 3786 in CD4⁺ T cells, 3048 in CD8⁺ T cells and 2179 in fibroblasts. The overlap of unaffected and affected skins in **Fig. 3a-d** (cluster 4) may represent an epigenetic signature of patients. It has also been reported that that clinically unaffected SSc skin has similar transcriptional changes as affected SSc skin compared to healthy skin¹⁸. So our findings would suggest that DCs are one of those cell types contributing to this molecular reprogramming before clinical disease is evident. Although DCs seem to have a much more significant enrichment in unaffected SSc skin (cluster 3 in **Fig. 3a-d**), there peaks barely enriched in any disease-relevant genomic features (**Graph 6d**), suggesting they are functionally irrelevant. On the other hand, peaks more accessible in affected skin were significantly enriched with SSc-relevant features from disease ontology analysis, further confirming the important role of DC in the pathogenesis of SSc.

Graph 6. Disease Ontologies of differential peaks in T cells, DC and fibroblasts. (a-d). Heatmap showing the enrichment of disease ontologies of peaks in Cluster1-6 in Fig. 3a-d. $-\log(P \text{ value})$ of enriched disease ontologies were shown.

14. In figure 3, “Interestingly, we noticed: (1) 28.5% of the differential peaks in fibroblast consist of peaks which represent a healthy cell state (cluster 2), but only 1.9% and 5.8% of these peaks were observed in T cells and DCs, respectively. This result was expected since fibrosis is the main distinction between clinically affected and unaffected skin.” This is hard to understand. First, do the authors mean that peaks for each cluster across immune cells are the same? From the plots, it doesn't look it that. Second, there are a large fraction of peaks are overlapped in unaffected skin and normal skin in cluster 2 for Fib. Second, what genes are enriched in the cluster 2 of Fib? Can the 28.5%

overlap allow you to draw the conclusion that “This result was expected since fibrosis is the main distinction between clinically affected and unaffected skin.”

Response: We agree with the reviewer that this statement is ambiguous. Peaks in cluster 2 were more accessible in normal and unaffected cells compare to affected cells, representing a healthy signature of cells. Our initial intention was to show that the epigenome of fibroblast in unaffected skin from SSc patients retain similarity with that of the healthy skin (**Graph 7**), while the epigenome of fibroblasts in clinically affected skin formed a distinct cluster. We have removed this ambiguous conclusion and extensively revised the description of the results in **Fig. 3** in the revised manuscript as follows:

“To further investigate the disease relevance and biological functions of these differential peaks, we performed disease and gene ontology analysis of all the peaks in each cluster for each cell type. We found: (1) Peaks in cluster 5, which were highly enriched in affected cells compared with normal and unaffected cells, representing an SSc disease signature. A number of autoimmune diseases, including SSc, were significantly more enriched in these peaks in DC (P value $\sim 10^{-14}$) compare with T cells and fibroblasts (P value ~ 1 , Supplementary Fig. 5, Supplementary Table 12), same as immune relevant biological functions (Supplementary Fig. 6), indicating a hidden epigenetic divergence in DCs that may be an underestimated factor in driving SSc. (2) Peaks in cluster 4 were more accessible in SSc patients compared with healthy donors, representing a patient signature. Disease associated biological functions such as “Cellular response to TGF β stimulus”, “ $\alpha\beta$ T cell activation”, “Inflammatory response” were found significantly enriched in cluster 4 peaks in T cells (P value $\sim 10^{-5}$, Supplementary Fig. 6a-b), suggesting that the chromatin states of the dermal T cells of SSc patients retain inherent abnormalities whether they are in the lesion or not.”

Graph 7. Cell type-specific regulome divergence in healthy, unaffected and affected skins (a). Pearson correlation of fibroblast (Fib) from normal, affected and unaffected cells.

15. I am concerned that the SSc samples represent a very small number of patients and that the authors have lumped samples from patients with limited cutaneous SSc and diffuse cutaneous SSc together. There is significant heterogeneity within these two clinical groups. To this end, the TF motifs show in Figure 3I for DCs seem to be enriched in one patient or another, but in only one case (NFkB) does it appear to be consistent across patients. This heterogeneity in the results needs to be explained within the context of the clinical and molecular subtypes. Are the DCs similarly enriched in both diffuse and limited SSc patient samples? Is there difference between the DCs of these two patient groups? In addition, figure 3L which quantifies the levels of ZBTB46 in SSc patients shows 5 patients with levels very similar to controls and 3 patients with high expression. This suggest significant variability of the DC population across SSc patients (see comment 12 above).

Response: We agree with the reviewer that there is significant variability of the DC population across SSc patients, and our analysis was based on both limited and diffuse cutaneous SSc patients. We did find that the ZBTB46 expression of both limited and diffuse SSc skin is significantly higher than that of normal skin (please see our response to comment 12 above), however, the small number of patients analyzed in this study prevent us from discussing the epigenomic heterogeneity of DCs from SSc patients. This limitation has been stated clearly in the **Discussion** of the revised manuscript.

16. The authors state on page 9, “in this way, we can systematically delineate all potential communications between T cells, DCs and Fibs in SSc”. This seems an over reach given the small number of SSc samples, and the variation among patients that are in this study. The number of patients examined here is vanishingly small. This statement should more clearly indicate this is a hypothesis generating exercise on a small number of patients.

Response: In **Fig. 4**, we described a computational model to explore the changed receptor-ligand interactions at affected SSc skin, and discovered many well-studied receptor-ligand signal pathways, such as the NOTCH and TGF β signal pathways, which fully supported the reliability of our data and analysis. Meanwhile, we also found several novel signaling pathways such as the EFNA5-EPHA4, HBEGF-CD44, XCL2-XCR1, CCL21-CCR7 interactions, which have not been reported in the published studies. Nevertheless, we agree with the reviewer that our result of cell-type crosstalk is a hypothesis generating exercise on a small number of patients and we have revised our expression of the conclusion as follows:

“In this way, we can nominate the potential communications between T cells, DCs and Fibs in SSc from limited number of patients’ samples.”

We also added a new paragraph in **Discussion** to highlight the limitation of our study, which provides a resource for hypothesis nomination, and should be tested by further functional studies in the future.

17. The conclusions of the paper need to clearly indicate the limitations of the study, in particular that other cell types, such as macrophages or cell types not isolated by their initial flow procedure (maybe some that are only now being discovered by single cell methods), can't be excluded based on this study.

Response: We agree and we have added final paragraph to the **Discussion** section about the limitations in this study. While comprehensive, our cell type specific analyses were not able to address all relevant cell types in SSc, in particular macrophages that are present in small numbers in normal skin, which preclude a comparison of macrophage epigenomic state in normal vs. SSc skin. Furthermore, cell types that are just now being recognized and not part of the flow-cytometry based prospective isolation were not studied. The knowledge generated in this work sets the stage for future efforts to address these outstanding and important questions.

Minor comments:

1. Suggest, changing the title of Fig. 1 to “Landscape of DNA accessibility in 8 cell types from normal skin *in vivo*”

Response: We have changed the title of **Fig. 1** to “Landscape of DNA accessibility in 8 cell types from normal skin *in vivo*”

2. How did you define the highest and lowest MRSS in Figure 2E. Please add this information to the figure legend for figure 2.

Response: We have added the definition of the highest and lowest mRSS in the revised figure legend for **Fig. 2e**.

3. Highlight the gene loci with more accessible in SSc in Fig. 4b.

Response: We have highlighted the gene loci with more accessible in SSc in the revised **Fig. 4b**.

4. Fig. 4b is hard to understand. Need a better interpretation in the Results part.

Response: We have added the following sentences in the revised Results to better interpret **Fig. 4b**:

“We then use a circos plot to show the ATAC-seq signals around selected receptors/ligands in normal versus affected skin from SSc patients, where the outermost torus displays the

names of the receptors and ligands in each cell type and the middle and inner torus display the ATAC-seq signals at these genes' loci in SSc patients and healthy controls respectively (Fig. 4b)"

5. In the methods, the publicly available dataset is mentioned as RNA-seq, when the data are DNA microarray based.

Response: We have corrected this issue in the revised **Methods**.

References:

- 1 Shin, J. Y. *et al.* Epigenetic activation and memory at a TGFB2 enhancer in systemic sclerosis. *Sci Transl Med* **11**, doi:10.1126/scitranslmed.aaw0790 (2019).
- 2 Wohlfahrt, T. *et al.* PU.1 controls fibroblast polarization and tissue fibrosis. *Nature* **566**, 344-349, doi:10.1038/s41586-019-0896-x (2019).
- 3 Maehara, T. *et al.* Cytotoxic CD4+ T lymphocytes may induce endothelial cell apoptosis in systemic sclerosis. *Journal of Clinical Investigation* **130**, 2451-2464, doi:10.1172/jci131700 (2020).
- 4 Tschumperlin, D. J., Liu, F. & Tager, A. M. Biomechanical regulation of mesenchymal cell function. *Curr Opin Rheumatol* **25**, 92-100, doi:10.1097/BOR.0b013e32835b13cd (2013).
- 5 Esensten, J. H., Helou, Y. A., Chopra, G., Weiss, A. & Bluestone, J. A. CD28 Costimulation: From Mechanism to Therapy. *Immunity* **44**, 973-988, doi:10.1016/j.immuni.2016.04.020 (2016).
- 6 Zaba, L. C., Fuentes-Duculan, J., Steinman, R. M., Krueger, J. G. & Lowes, M. A. Normal human dermis contains distinct populations of CD11c(+)BDCA-1(+) dendritic cells and CD163(+)FXIII(+) macrophages. *Journal of Clinical Investigation* **117**, 2517-2525, doi:10.1172/Jci32282 (2007).
- 7 Fuentes-Duculan, J. *et al.* A Subpopulation of CD163-Positive Macrophages Is Classically Activated in Psoriasis. *Journal of Investigative Dermatology* **130**, 2412-2422, doi:10.1038/jid.2010.165 (2010).
- 8 Rodero, M. P. & Khosrotehrani, K. Skin wound healing modulation by macrophages. *Int J Clin Exp Pathol* **3**, 643-653 (2010).
- 9 Mahdavian Delavary, B., van der Veer, W. M., van Egmond, M., Niessen, F. B. & Beelen, R. H. Macrophages in skin injury and repair. *Immunobiology* **216**, 753-762, doi:10.1016/j.imbio.2011.01.001 (2011).
- 10 Hinchcliff, M. *et al.* Mycophenolate Mofetil Treatment of Systemic Sclerosis Reduces Myeloid Cell Numbers and Attenuates the Inflammatory Gene Signature in Skin. *J Invest Dermatol* **138**, 1301-1310, doi:10.1016/j.jid.2018.01.006 (2018).
- 11 Furst, D. E. *et al.* The modified Rodnan skin score is an accurate reflection of skin biopsy thickness in systemic sclerosis. *J Rheumatol* **25**, 84-88 (1998).
- 12 Satpathy, A. T. *et al.* Zbtb46 expression distinguishes classical dendritic cells and their committed progenitors from other immune lineages. *J Exp Med* **209**, 1135-1152, doi:10.1084/jem.20120030 (2012).
- 13 Skaug, B. *et al.* Global skin gene expression analysis of early diffuse cutaneous systemic sclerosis shows a prominent innate and adaptive inflammatory profile. *Annals of the Rheumatic Diseases* **79**, 379-386, doi:10.1136/annrheumdis-2019-215894 (2020).
- 14 Assassi, S. *et al.* Dissecting the heterogeneity of skin gene expression patterns in systemic sclerosis. *Arthritis Rheumatol* **67**, 3016-3026, doi:10.1002/art.39289 (2015).
- 15 Hamada, M., Tsunakawa, Y., Jeon, H., Yadav, M. K. & Takahashi, S. Role of MafB in macrophages. *Exp Anim Tokyo* **69**, 1-10, doi:10.1538/expanim.19-0076 (2020).
- 16 Goudot, C. *et al.* Aryl Hydrocarbon Receptor Controls Monocyte Differentiation into Dendritic Cells versus Macrophages. *Immunity* **47**, 582+, doi:10.1016/j.immuni.2017.08.016 (2017).

- 17 Satpathy, A. T., Wu, X. D., Albring, J. C. & Murphy, K. M. Re(de)fining the dendritic cell lineage. *Nat Immunol* **13**, 1145-1154, doi:10.1038/ni.2467 (2012).
- 18 Whitfield, M. L. *et al.* Systemic and cell type-specific gene expression patterns in scleroderma skin. *P Natl Acad Sci USA* **100**, 12319-12324, doi:10.1073/pnas.1635114100 (2003).

REVIEWERS' COMMENTS

Reviewer #2 (Remarks to the Author):

The authors have addressed points raised in my review and the paper is now substantially improved.

Reviewer #3 (Remarks to the Author):

I appreciate the authors detailed efforts to address the suggestions raised in my initial review. I would ask that figures provided in the rebuttal be added to the supplementary materials and referenced appropriately in the text. It appears that Graphs 1, 5c and 6 are included but Graphs 2 – 5a,b, and Graph 7 do not appear to be included in the main figures or the supplement.

Responses to reviewers

We thank the reviewers for their positive assessment and thoughtful suggestions that have improved this work. Point-by-point responses are as follows.

Reviewer #2 (Remarks to the Author):

The authors have addressed points raised in my review and the paper is now substantially improved.

Response: Thank you for the positive feedback to our manuscript.

Reviewer #3 (Remarks to the Author):

I appreciate the authors detailed efforts to address the suggestions raised in my initial review. I would ask that figures provided in the rebuttal be added to the supplementary materials and referenced appropriately in the text. It appears that Graphs 1, 5c and 6 are included but Graphs 2 – 5a,b, and Graph 7 do not appear to be included in the main figures or the supplement.

Response: Thank you for the positive feedback to our manuscript. We have added the relevant figures in the rebuttal letter to the supplementary materials as follows, which were referenced appropriately. We have also updated the figure legends accordingly.

- (1) Graph 2 as the revised Supplementary Fig. 1b
- (2) Graph 3a-b as the revised Supplementary Fig. 3a-b
- (3) Graph 3c as the revised Supplementary Fig. 4
- (4) Graph 4a as the revised Supplementary Fig. 6a
- (5) Graph 5a as the revised Supplementary Fig. 11

However, since Graph4b, Graph5c and Graph7 serve only as the supporting evidences for the previous revision in response to the reviewer's questions, and these results were not mentioned in the revised manuscript, we therefore decide not to include them as supplementary figures in the current revision.